# Towards Effective Evaluations and Comparisons for LLM Unlearning Methods

**Qizhou Wang**[1*]  **Bo Han**[1,2†]  **Puning Yang**[1]  **Jianing Zhu**[1]
**Tongliang Liu**[3]  **Masashi Sugiyama**[2,4]

[1]TMLR Group, Department of Computer Science, Hong Kong Baptist University
[2]RIKEN Center for Advanced Intelligence Project
[3]Sydney AI Center, The University of Sydney
[4]The University of Tokyo

## Abstract

The imperative to eliminate undesirable data memorization underscores the significance of machine unlearning for large language models (LLMs). Recent research has introduced a series of promising unlearning methods, notably boosting the practical significance of the field. Nevertheless, adopting a proper evaluation framework to reflect the true unlearning efficacy is also essential yet has not received adequate attention. This paper seeks to refine the evaluation of LLM unlearning by addressing two key challenges—**a)** the robustness of evaluation metrics and **b)** the trade-offs between competing goals. The first challenge stems from findings that current metrics are susceptible to various red teaming scenarios. It indicates that they may not reflect the true extent of knowledge retained by LLMs but rather tend to mirror superficial model behaviors, thus prone to attacks. We address this issue by devising and assessing a series of candidate metrics, selecting the most robust ones under various types of attacks. The second challenge arises from the conflicting goals of eliminating unwanted knowledge while retaining those of others. This trade-off between unlearning and retention often fails to conform the Pareto frontier, rendering it subtle to compare the efficacy between methods that excel only in either unlearning or retention. We handle this issue by proposing a calibration method that can restore the original performance on non-targeted data after unlearning, thereby allowing us to focus exclusively on assessing the strength of unlearning. Our evaluation framework notably enhances the effectiveness when assessing and comparing various LLM unlearning methods, further allowing us to benchmark existing works, identify their proper hyper-parameters, and explore new tricks to enhance their practical efficacy. The code is publicly available at: https://github.com/tmlr-group/Unlearning-with-Control.

## 1 Introduction

Large language models (LLMs), like Llama (Touvron et al., 2023a;b) and GPT (Brown et al., 2020; Achiam et al., 2023), have exhibited remarkable proficiency in general-purpose language generation and understanding (Azerbayev et al., 2023; Roziere et al., 2023; Wu et al., 2023; Thirunavukarasu et al., 2023; Zhou et al., 2024; Huang et al., 2024). These advancements are credited to the development of Transformer-based architectures (Vaswani et al., 2017) with billions of parameters and to the extensive pre-training on web-sourced corpora with trillions of tokens (Brown et al., 2020). However, on the other side, scaling up models aggravates the risk of memorizing effects (Arpit et al., 2017) and sourcing from the web makes LLMs inherent its inaccuracies and biases (Liu et al., 2023a). It raises the invoking concerns for LLM privacy and fidelity, posing a long array of undesirable LLM behaviors sourced from training corpora (Liu et al., 2023a), including copyright (Yao et al., 2023a), fairness (Gallegos et al., 2023), and toxicity (Liu et al., 2023b), among many others.

---

[*]Work done during internship at RIKEN Center for Advanced Intelligence Project.
[†]Correspondence to Bo Han (bhanml@comp.hkbu.edu.hk).

**How to Erase Undesirable Data Memorization in LLMs?** Machine unlearning (Bourtoule et al., 2021; Zhu et al., 2024) offers a general solution. In the context of LLMs, the primary goal of unlearning is to precisely remove the parameterized knowledge related to unlearning targets meanwhile maintaining model performance for non-targets (Liu et al., 2024). The unlearning targets within LLMs are typically characterized by an unlearning set, denoted as $\mathcal{D}_u = \{s_u = [x, y_u]\}_{n_u}$, and we need to develop unlearning methods upon $\mathcal{D}_u$ that meet the goals of LLM unlearning. Some of the noteworthy baselines are gradient ascent (GA) (Yao et al., 2023b), gradient difference (GD) (Maini et al., 2024), and negative preference optimization (NPO) (Zhang et al., 2024).

While algorithmic designs are crucial, their proper evaluations are equally vital. Misleading metrics can lead us to overestimate the unlearning efficacy, potentially causing severe consequences when applying these methods in practice. In general, effective unlearning metrics should accurately quantify the extent of knowledge parametrization. Previous studies have introduced a set of intriguing metrics, such as "familiarity" (Eldan & Russinovich, 2023), "model utility" (Maini et al., 2024), "forget quality" (Maini et al., 2024), and "QA accuracy" (Li et al., 2024). However, these metrics are often intertwined or reliant on manual-designed prompting, which are not general. Even worse, recent works (Lynch et al., 2024) have shown that some metrics are highly susceptible to various red teaming attacks, such as jail-breaking (Shen et al., 2023). It indicates that the current metrics might not adequately reflect the extent to which targeted knowledge is erased—even if models notably retain the targeted knowledge, these metrics may still falsely indicate its complete removal.

We conjecture that an effective metric for unlearning should exhibit robustness across diverse red teaming scenarios. This robustness can manifest as strong linear correlations between metric scores calculated from the original unlearning set and those computed after attacking. Large distortion in this correlation would suggest that the associated metrics fail to capture the extent of knowledge parametrization, instead mirroring more superficial behaviors that are vulnerable to attacks. To investigate effective metrics for LLM unlearning, we consider a set of basic metrics, either derived from previous works or mentioned in other related fields (Duan et al., 2024), cf., Section 3. We further examine their robustness under four red teaming behaviors, including jail-breaking (Shen et al., 2023), embedding probing (Belrose et al., 2023), relearning (Lo et al., 2024), and token noising. Then, measuring by the Pearson correlation coefficient (PCC) (Cohen et al., 2009), we observe that the extraction strength (ES)—quantifying the amount of information required to recover original outputs—emerges to be the most effective choice, thus employed for assessing unlearning.

Even with the ES as an effective metric, comparing various LLM unlearning methods remains a challenging issue. This difficulty primarily arises from the need to balance between two conflicting goals for effective unlearning: retaining performance on non-targeted data (**retention**) and removing targeted knowledge (**removal**). For example, when comparing two unlearned models, it is common the case where one model outperforms in removal but another one excels at retention, making it difficult to determine which one is overall superior, cf., Figure 1. We address this issue by aligning their common performance, i.e., their capacity of retention, in a post-unlearning manner. Motivated by (Wortsman et al., 2022), it is achieved by mixing model parameters from both before and after unlearning, modulated through a mixing factor $\alpha$. With proper control via $\alpha$, we observe that model mixing (MM) enables us to finely calibrate the extent of unlearning such that performance on common data is adequately preserved, meanwhile the inevitable compromise on the extent of removal is roughly minimized, cf., Section 4. Thereafter, we can fairly concentrate on assessing the strength of the removal on

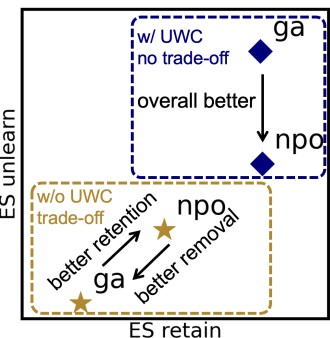

Figure 1: For effective unlearning, it is preferable to have large ES scores for retention (**x-axis**) yet small for removal (**y-axis**). For the raw results (orange), we observe that GA excels at removal whereas NPO is better in retention, making it hard to determine which method is overall better. UWC resolves this challenge by aligning ES scores for retention, allowing us to focus on comparing the ES scores for unlearning (blue). It leads to the conclusion that NPO is overall superior.

targeted data, thereby alleviating the challenges for comparing different unlearning methods or unlearned models when pursuing to goals of removal and retention concurrently.

We refer to our evaluation framework as "unlearning with control" (UWC), which incorporates the ES as the basic metric and utilizes MM for calibration to ease assessments and comparisons across methods/setups. Based on UWC, we benchmark a series of representative works along with suggestions for their hyper-parameters. We challenge the currently perceived advancements in LLM unlearning, where the ostensibly positive behaviors of current state-of-the-art methods may be the result of either excessive unlearning or insufficient unlearning. Nevertheless, proper hyper-parameter tuning can remarkably enhance the efficacy of many earlier works, such as GA variants, showing potential to exceed many advanced counterparts. Leveraging UWC, we also benefit the community by exploring a range of simple yet intriguing tricks to further enhance the practical efficacy of current unlearning methods, which are not covered in previous works.

## 2 LLM Learning and Unlearning

To begin with, we discuss the necessary backgrounds for LLM learning as well as unlearning.

**LLM Learning.** We study the LLM parameterized by $\boldsymbol{\theta}$ with layer-wise self-attention structures (Liu et al., 2018). Upon receiving an input $s$, the LLM estimates the probability distributions, denoted by $p(\cdot|s;\boldsymbol{\theta})$, over the next possible tokens. The LLM is trained on a substantial web-scale corpora, denoted by $\mathcal{D}_{\mathrm{t}} = \{s = [x, y]\}_{n_{\mathrm{t}}}$ of size $n_{\mathrm{t}}$. During training, we aim at minimizing the prediction loss $\ell(y|x;\boldsymbol{\theta}) = -\log p(y|x;\boldsymbol{\theta})$ over $\mathcal{D}_{\mathrm{t}}$. The resulting LLM is capable of properly handling a wide range of language generation tasks. We adopt the notation $y^i$ to represent the $i$-th token, $y^{<i}$ for the prefix up to the $i$-th token, and the string generated via greedy decoding by $f(s;\boldsymbol{\theta})$.

**LLM Unlearning.** However, employing training corpora sourced from the wild heavily raises the risk that our LLMs will learn from sensitive information, thereby precipitating a host of legal and ethical concerns (Yao et al., 2023a; Ji et al., 2023; Gallegos et al., 2023; Liu et al., 2023b).These issues further necessitate the need for a post-training mechanism that enables our LLMs to eradicate any associated parameterized knowledge that is undesirable. This requirement motivates the recent research on LLM unlearning (Yao et al., 2023b; Maini et al., 2024), formalizing the above goal by involving so-called the unlearning set $\mathcal{D}_{\mathrm{u}} = \{s_{\mathrm{u}} = [x, y_{\mathrm{u}}]\}_{n_{\mathrm{u}}}$ ($n_{\mathrm{u}} \ll n_{\mathrm{t}}$, typically). Overall, LLM unlearning aims to adjust model parameters $\boldsymbol{\theta}$ such that the content related to $\mathcal{D}_{\mathrm{u}}$ is erased. More specifically, for practical-effective unlearning, it should pursue two goals simultaneously:

- **Removal**: The knowledge associated with the unlearning dataset $\mathcal{D}_{\mathrm{u}}$ should notably deteriorate, revealing effective unlearning on parametrization that targeted to be erased.

- **Retention**: The knowledge for other data, following $\mathcal{D}_{\mathrm{t}} \backslash \mathcal{D}_{\mathrm{u}}$, should be retained, such that common model responses are sufficiently preserved, thereby ensuring its overall integrity.

To ease our discussion below, we distinguish between two types of data: **a)** targeted data, which are targeted to be unlearned (i.e., within the unlearning set $\mathcal{D}_{\mathrm{u}}$), and **b)** non-targeted data, which are required to be retained (i.e., all other data within $\mathcal{D}_{\mathrm{t}} \backslash \mathcal{D}_{\mathrm{u}}$). Moreover, for the generalization perspective of unlearning, we aim for the unlearned models to not recall the targeted knowledge by assessing on a rephrased version of $\mathcal{D}_{\mathrm{u}}$, adhering to the standard setup as in (Maini et al., 2024).

**Unlearning Methods.** Stemming from formalization for the above two goals, gradient difference (GD) (Maini et al., 2024) has established as a foundational baseline. Its unlearning objective is

$$-\underbrace{\mathbb{E}_{s_{\mathrm{u}} \sim \mathcal{D}_{\mathrm{u}}} \ell\big(y_{\mathrm{u}}|x;\boldsymbol{\theta}\big)}_{\text{unlearning risk}} + \lambda \underbrace{\mathbb{E}_{s \sim \mathcal{D}_{\mathrm{t}} \backslash \mathcal{D}_{\mathrm{u}}} \ell\big(y|x;\boldsymbol{\theta}\big)}_{\text{retaining risk}}, \tag{1}$$

which composes of two terms: the unlearning risk and the retaining risks, balanced by the hyper-parameter $\lambda$. The unlearning risk increases the prediction losses for undesirable responses $y_{\mathrm{u}}$, aligning with gradient ascent (GA) when updating LLMs. The retaining risk is implemented to retain the original model integrity, aiming to ensure that the responses for non-targeted data remain unchanged. Despite its mechanisms, previous works believe that GD is still susceptible to catastrophic collapse (Zhang et al., 2024), wherein LLM parameters are remarkably altered and common model responses are severely distorted after unlearning. To further enhance the practical utility, a series

of subsequent works have been explored. Among them, methods such as KL (Maini et al., 2024), NPO (Zhang et al., 2024), PO (Maini et al., 2024), and RMU (Li et al., 2024), are well-established and have received reasonable attentions. Please refer to Appendix C for more discussions.

## 3    EVALUATION METRICS

Accompanying advances made in algorithmic designs, it is also essential to accurately assess the effectiveness for various unlearning methods. Particularly, an inappropriate evaluation framework, such as those that overestimate the strength of unlearning, can mislead practitioners to be overconfident on the reliability of the resulting unlearned models. An ideal evaluation framework for LLM unlearning should effectively quantify the extent to which targeted knowledge remains parameterized within. Moreover, it should be general-actionable across tasks, simply to implement, and free from specific prompt engineering that may introduce modeling and prompting bias.

In our pursuit of such an evaluation framework, we begin by examining a series of basic metrics to determine their robustness and suitability, as detailed in the following.

- **Perplexity** (PPL) (Chang et al., 2024): assessing the model confidence of auto-regressive models, defined as the exponentiation of the cross entropy, i.e., $\exp\{-\log p(y|x; \boldsymbol{\theta})\}$.

- **ROUGE-L** (ROUGE) (Lin, 2004): measuring output quality by the proportion of the longest common sub-sequence presents between the ground truth $y$ and the model response $f(x; \boldsymbol{\theta})$.

- **Exact Memorization** (EM) (Tirumala et al., 2022): measuring output quality by the proportion of the same tokens with the ground truth $y$, i.e., $\frac{1}{|y|} \sum_k \mathbf{1}\{\arg\max_y f(y|[x, y^{<k}]; \boldsymbol{\theta}) = y^k\}$, where $\mathbf{1}\{\cdot\}$ returns 1 if the condition therein is true, otherwise 0.

- **Extraction Strength** (ES) (Carlini et al., 2021): quantifying the strength of memorization by the minimal proportion of the prefix to recover the suffix. To better align with its name, we adjust the metric to use 1 minus its negative value, i.e., $1 - \frac{1}{|y|} \min_k \{k | f([x, y^{<k}]; \boldsymbol{\theta}) = y^{>k}\}$.

- **KL Divergence** (KL): the KL divergence for predictions between original and unlearned models. It is formalized as $\mathrm{KL}\big[p(y|x; \boldsymbol{\theta}) \, || \, p(y|x; \boldsymbol{\theta}_{\mathrm{ref}})\big]$ with $\mathrm{KL}$ the operation of the KL divergence.

These metrics cover a broad range of practical metrics that are widely recognized in prior research. For example, PPL is used as a part of the metrics for "model utility" in (Maini et al., 2024), and the "rewrite score" in (Patil et al., 2023), among many others (Patil et al., 2023); EM serves as the key metric for (Barbulescu & Triantafillou, 2024; Jin et al., 2024); ROUGE is adopted in (Du et al., 2024; Maini et al., 2024); KL is mentioned in (Garg et al., 2024). We also take into account a less common yet intriguing metric that quantifies data memorization, i.e., ES, particularly pertinent in studies of membership attacks (Garg et al., 2024). Nevertheless, we exclude certain metrics that are difficult to compute, such as those dependent on *gold standard* models that require the full re-training without targeted data (Garg et al., 2024; Thudi et al., 2022; Maini et al., 2024). Moreover, for generality, we also disregard task-specific metrics, including GPT-based evaluations (Lynch et al., 2024; Eldan & Russinovich, 2023), QA accuracy that relies on manual-designed multiple choice questions (Patil et al., 2023; Li et al., 2024), and those dependent on task-specific detectors (Yao et al., 2023b).

**What Ensures a Good Metric?** Among candidates, we wonder whether they can effectively quantify the internal parametrization of knowledge, a question that is directly tied to the general goals of LLM unlearning, as mentioned in Section 2. Overall, a proper metric should demonstrate robustness against various red teaming scenarios; if not, it risks only capturing superficial model behaviors, thereby vulnerable to manipulative attacks (cf., Appendix A). To gauge this robustness, we examine the metrics with several representative attacking behaviors considered in the following.

- **Jail-breaking** (Shen et al., 2023): manipulating LLM behaviors to elicit undesirable knowledge via crafted prompts. A proper metric should be robust to jail-breaking attacks.

- **Probing** (Belrose et al., 2023): decoding middle embeddings via extra linear unembedding modules. It should be hard to recover unlearned knowledge from embeddings after proper unlearning.

- **Relearning** (Lo et al., 2024): few-shot fine-tuning for unlearned LLMs. In an ideal case, unlearned models are hard to sufficiently relearn the previously unlearned knowledge.

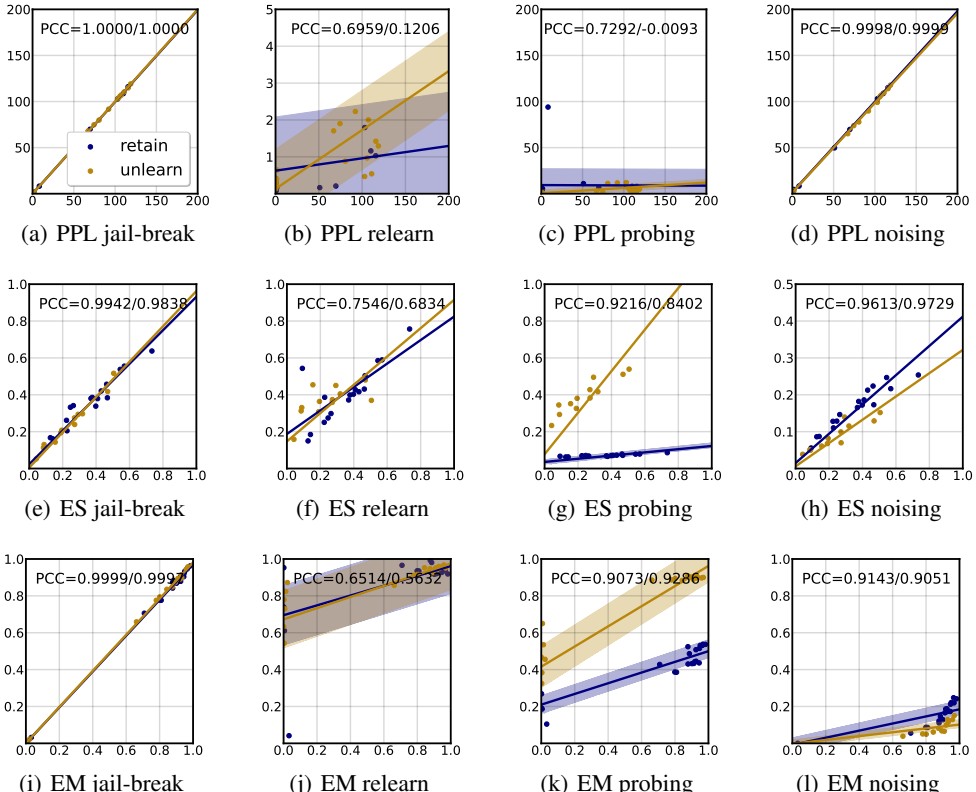

Figure 2: **Metric Robustness under Red Teaming Attacks.** We depict the metric scores before (**x-axis**) and after (**y-axis**) attacks jointly for different unlearning setups: across 2 LLMs (Phi-1.5 and Llama-2-7B), 3 unlearning percentages (1%, 5%, and 10%), and 4 unlearning methods (GA, GD, PO, and NPO). We consider 3 representative metrics under 4 red teaming behaviors. We apply the log-scale for PPL to avoid numeric errors. For each of these scenarios, we compute the PPC with respect to targeted and non-targeted data respectively, displayed at the top of each figure (targeted data / non-targeted data). We provide linear fits for targeted and non-targeted data separately, accompanied by shaded areas representing the standard deviations that visualize the PPC scores.

- **Token Noising**: perturbing 5% of tokens within each $s$ by replacing them with random tokens. The resulting strings with token noise are used as targets when computing scores across metrics.

Some attacking scenarios have been explored in previous works (Lynch et al., 2024), such as relearning and jail-breaking, while others, like probing and token noise, remain less explored. These four attacking scenarios are motivated by a broader interest in comprehending LLM behaviors across diverse contexts. For example, LLMs may maintain knowledge without explicitly outputting it (Patil et al., 2023), a phenomenon related to jail-breaking; parameterized knowledge can be extracted from embeddings (Belrose et al., 2023), pertaining to probing attacks; fine-tuning may inadvertently lead to emergence of harmful model behaviors (Lo et al., 2024), associated with relearning. Please refer to Appendix D for detailed descriptions on these attacking strategies. Also, as discussed in Appendix A, jail-breaking and probing are more important for assessing robustness than other ones.

**How to Assess the Metric Robustness?** To account for the inherent challenges posed for varying attacks aforementioned, it is generally unrealistic to expect that the metric scores to remain unchanged. A more reasonable, yet still rigorous, criterion is to examine whether the metrics exhibit a linear relationship between the original values and that after attacks. Accordingly, although values may change, the relative rankings (i.e., the orders of superiority across unlearned models) remains the same without skewing. Please refer to Appendix A for a more formal discussion. Here, we use Pearson correlation coefficient (PCC) (Cohen et al., 2009) to gauge the linear correlation before

and after attacks. Note that the potential sensitivities could be attributed to either the limitations of metrics or unlearning methods, yet distinguishing between these two factors is hard. We mitigate this issue by computing the PCC across LLMs, unlearning setups, and various unlearning methods, neutralizing influences from those factors unrelated to the metrics themselves to much extent.

**Results.** Due to space limit, we examine the robustness of three representative metrics among five candidate metrics, across various attacks as illustrated in Figure 2, please refer to Section 6 for the experimental setups and Appendix D for more results. We observe that relearning has the largest impacts on the robustness of metrics, mainly due to the further tuning of parameters for unlearned LLMs. Under relearning attacks, ROUGE shows to be the least effective metric, while ES is our best choice. The probing attacks also have substantial impacts, particularly on the PPL for non-targeted data, even demonstrating negative correlations. Under probing attacks, the ES is more robust than other candidates. At last, jail-breaking and feature noising attacks are generally less effective at disturbing the metrics, with ROUGE again exhibiting the least robustness. Overall, ES stands out as the most reliable metric for LLM unlearning. It shows superior robustness during relearning and probing attacks, and maintains a small PCC gap over the PPL for other attacks.

**The ES Metrics for Assessing Unlearning.** Based on our evaluations above, we recommend ES as our proper choice for assessing the extent of parameterized knowledge. It is versatile across various unlearning setups and can properly quantify unlearning behaviors with respect to both removal and retention. For removal, the average ES, calculated for targeted data as

$$\mathrm{ES}(\mathcal{D}_\mathrm{u};\boldsymbol{\theta}) = \mathbb{E}_{(x,y_\mathrm{u})\sim D_\mathrm{u}}\big[1 - \frac{1}{|y_\mathrm{u}|}\min_k\{k|f([x,y_\mathrm{u}^{<k}];\boldsymbol{\theta}) = y_\mathrm{u}^{>k}\}\big], \tag{2}$$

should be **small** after unlearning. For retention, the average ES for non-targeted data should be **high**:

$$\mathrm{ES}(\mathcal{D}_\mathrm{t}\backslash\mathcal{D}_\mathrm{u};\boldsymbol{\theta}) = \mathbb{E}_{(x,y)\sim\mathcal{D}_\mathrm{t}\backslash\mathcal{D}_\mathrm{u}}\big[1 - \frac{1}{|y|}\min_k\{k|f([x,y^{<k}];\boldsymbol{\theta}) = y^{>k}\}\big]. \tag{3}$$

ES will be used as the basic metric for evaluating LLM unlearning in our experiments below.

## 4 FAIR COMPARISON

An essential aspect of quantifying unlearning performance is enabling their reliable comparison, which can facilitate the identification of superior unlearning methods and effective hyper-parameter configurations. However, achieving such a fair comparison is not straightforward for unlearning, even with the ES as an effective metric. The challenge mainly originates from the inherent trade-off between removal and retention, both of which are crucial for unlearning efficacy.

Often, unlearning methods that excel at removing targeted data will under-perform in retaining non-targeted knowledge, and vice versa. This scenario necessitates subjective judgments to balance their trade-offs and identify the overall superior choice. Figure 1 presents an example: When comparing between NPO and GA, we observe that the ES computed on targeted data for GA is smaller than that for NPO, indicating GA is more effective in erasing targeted knowledge. On the other side, the ES computed on non-targeted data for NPO is higher than that for GA, suggesting that NPO better preserves the original model performance. While GA may be the appropriate choice when focusing solely on removal, its efficacy relative to NPO becomes less clear when retention is also considered. This scenario is commonly observed in existing methods, cf., Section 6, where their claimed improvements often do not align with the Pareto frontiers between removal and retention.

**On the Importance of Calibration.** To ensure an easy and fair way of comparison, our motivation is to align LLM performance on non-targeted data post-unlearning, i.e., aligning the ES scores on non-targeted data across methods. Once this calibration can be established, we can focus solely on the ES comparison on targeted data. Refer to Figure 1 for the illustration. To achieve the goal of proper calibration, we seek for a flexible control method that permits the adjustment for the extent of unlearning after the unlearning procedure. Inspired by parameter disentanglement (Wortsman et al., 2022; Ilharco et al., 2022)—where mixing parameters from two models can endow the resulting one with characteristics from both, akin to model ensemble (Ortiz-Jimenez et al., 2023)—we propose **model mixing** (MM) as a flexible method for such control. Formally, considering parameters before unlearning, denoted as $\boldsymbol{\theta}_\mathrm{ref}$, and after unlearning, denoted as $\boldsymbol{\theta}$, their mixture is given by

$$(1-\alpha)\boldsymbol{\theta}_\mathrm{ref} + \alpha\boldsymbol{\theta}, \tag{4}$$

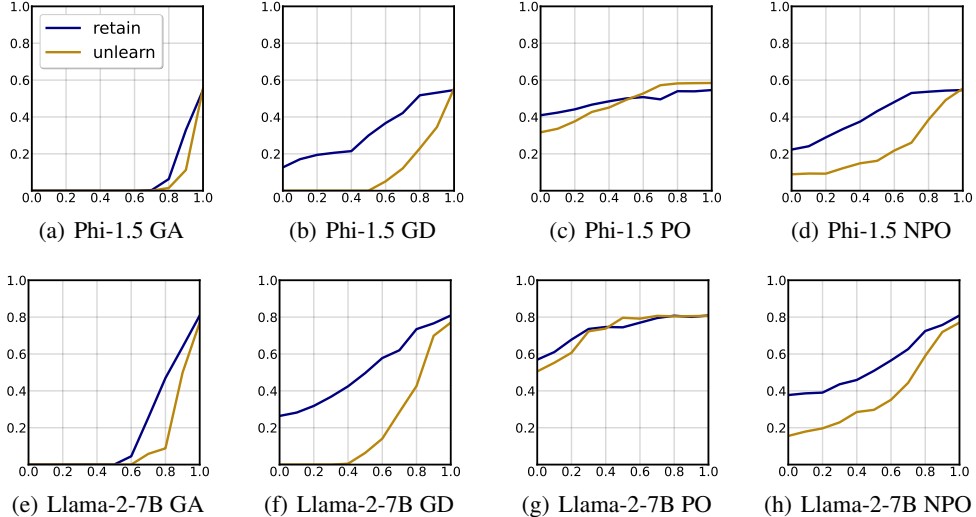

Figure 3: **ES Scores with MM Control.** We depict values of $\alpha$ (**x-axis**) versus the ES scores (**y-axis**) on targeted (unlearn) and non-targeted (retain) data. We consider 2 LLMs (Phi-1.5 and Llama-2-7B) and 4 unlearning methods (GA, GD, PO, and NPO) under the 5% TOFU unlearning setup.

with $0 \leq \alpha \leq 1$ the mixing factor that should be searched. In general, a lower $\alpha$ emphasizes the parametrization of the original model, whereas a higher $\alpha$ accentuates those of the unlearned one. By careful-adjusted $\alpha$, we can control the extent of unlearning to align performance on non-targeted data, such that the associated ES scores can be maintained, e.g., similar to those before unlearning.

**Is MM Proper for Calibration?** The answer is YES. We observe that MM ensures a smooth control over the extent of unlearning, supported by an overall monotonic relationship between $\alpha$ and the ES scores. We illustrate several examples in Figure 3 as evidence of this effect. The benefits of this smooth control extend beyond stability, which enabling the calibration of unlearned models such that the strength of removal on targeted data is minimally compromised. Therefore, comparisons of ES scores on targeted data after calibration are fair and valid. This smooth control also facilitates us to suggest an efficient method for the estimation of the optimal $\alpha$, as detailed in Appendix E.

At first glance, it seems that hyper-parameter tuning can also be used for calibration. To highlight the superiority of MM, we would like to emphasize that a proper calibration method should ensure the control is applied in a noticeable yet smooth manner. However, as observed in Appendix H, the model behaviors are quite sensitive to the choices of hyper-parameters, and we often do not achieve the desired level of recovery even with intensive tuning. In contrast, in Figure 3, the control exerted by MM over model behaviors is smooth. Additionally, conducting calibration through hyper-parameter tuning is too method-specific, and its computational costs are also prohibitively high. By contrast, MM can be applied post-unlearning across different methods without incurring the additional costs associated with re-unlearning. Therefore, we conclude that MM is more general, reliable, flexible, and efficient than hyper-parameter tuning in calibration.

## 5 UNLEARNING WITH CONTROL

With the ES as the basic metric and the MM for performance calibration, we name the overall framework as **unlearning with control** (UWC). It is a two-step evaluation strategy, consisting of **a)** calibration and **b)** assessment, structured in the following.

- **Calibration**: We control the extent of unlearning such that the ES scores on non-targeted data should be close to that before unlearning. Formally, we aim for the largest possible $\alpha$ such that at least $\tau \times 100\%$ of the original ES scores on non-targeted data can be preserved, i.e.,

$$\max_{\alpha} \left\{ \alpha \mid \mathrm{ES}(\mathcal{D}_\mathrm{t} \backslash \mathcal{D}_\mathrm{u}; (1-\alpha)\boldsymbol{\theta}_\mathrm{ref} + \alpha\boldsymbol{\theta}) > \tau \mathrm{ES}(\mathcal{D}_\mathrm{t} \backslash \mathcal{D}_\mathrm{u}; \boldsymbol{\theta}_\mathrm{ref}) \right\}, \tag{5}$$

Table 1: Comparison between different unlearning methods on TOFU fictitious unlearning with UWC calibration. ↓ / ↑ indicate smaller / larger values are preferable. We primarily focus on the ES scores for unlearning (shaded), given that the ES scores for retention are calibrated.

| | LLM | Phi-1.5 | | | | Llama-2-7B | | | |
|---|---|---|---|---|---|---|---|---|---|
| | | ES-exact | | ES-perturb | | ES-exact | | ES-perturb | |
| setup | method | retain ↑ | unlearn ↓ | retain ↑ | unlearn ↓ | retain ↑ | unlearn ↓ | retain ↑ | unlearn ↓ |
| 1% | before unlearning | 0.4433 | 0.5969 | 0.2115 | 0.1605 | 0.8277 | 0.8039 | 0.5302 | 0.4001 |
| | GA | 0.4262 | 0.3748 | 0.2071 | 0.1551 | 0.7536 | 0.1333 | 0.4976 | **0.0230** |
| | GD | 0.4212 | 0.3449 | 0.2072 | 0.1413 | 0.7471 | **0.0293** | 0.4471 | 0.1860 |
| | KL | 0.4232 | 0.2123 | 0.2005 | 0.0840 | 0.7337 | 0.0515 | 0.4428 | 0.0913 |
| | PO | 0.4242 | 0.6001 | 0.1936 | 0.1468 | 0.7508 | 0.2387 | 0.4757 | 0.2509 |
| | NPO | 0.4424 | **0.1259** | 0.2136 | **0.0702** | 0.7383 | 0.2543 | 0.4776 | 0.1703 |
| | RMU | 0.4245 | 0.4682 | 0.2115 | 0.1855 | 0.7559 | 0.5093 | 0.4096 | 0.3538 |
| 5% | before unlearning | 0.4433 | 0.5619 | 0.2115 | 0.2374 | 0.8277 | 0.7735 | 0.5302 | 0.4126 |
| | GA | 0.4497 | **0.2958** | 0.2136 | 0.2349 | 0.7780 | 0.7033 | 0.4031 | 0.4765 |
| | GD | 0.3919 | 0.4140 | 0.2004 | **0.0045** | 0.7432 | 0.3385 | 0.4775 | 0.3166 |
| | KL | 0.3823 | 0.3766 | 0.1794 | 0.1614 | 0.7207 | **0.0953** | 0.4814 | **0.1516** |
| | PO | 0.4086 | 0.4524 | 0.2020 | 0.2343 | 0.7715 | 0.5496 | 0.4792 | 0.3502 |
| | NPO | 0.4433 | 0.3768 | 0.1836 | 0.1509 | 0.7207 | 0.1104 | 0.4804 | 0.2777 |
| | RMU | 0.4404 | 0.4252 | 0.2047 | 0.2147 | 0.7112 | 0.4034 | 0.4927 | 0.3884 |
| 10% | before unlearning | 0.4433 | 0.5299 | 0.2115 | 0.1843 | 0.8277 | 0.8307 | 0.5302 | 0.3099 |
| | GA | 0.3796 | **0.2486** | 0.2137 | 0.1624 | 0.7015 | 0.4916 | 0.4825 | 0.2419 |
| | GD | 0.4454 | 0.4935 | 0.1761 | **0.0345** | 0.7771 | **0.0980** | 0.4780 | **0.1200** |
| | KL | 0.4424 | 0.4912 | 0.2075 | 0.0922 | 0.7765 | 0.2791 | 0.4734 | 0.1236 |
| | PO | 0.4177 | 0.5499 | 0.2042 | 0.1786 | 0.7543 | 0.7397 | 0.5302 | 0.3435 |
| | NPO | 0.4072 | 0.3499 | 0.2028 | 0.1281 | 0.7769 | 0.3700 | 0.5100 | 0.1243 |
| | RMU | 0.4364 | 0.5208 | 0.1944 | 0.1547 | 0.7874 | 0.7526 | 0.4871 | 0.3196 |

where $\tau$ should be close to 1 to ensure strong calibration. Note that we pursue for the largest $\alpha$ to minimize the compromise on the strength of removal, as mentioned in Section 4.

- **Assessment**: For unlearned LLMs that are well calibrated for retention, one can fairly evaluate and compare their strength of removal, i.e., their ability to erase parameterized knowledge targeted to be unlearned. The overall efficacy of unlearning can then be accurately assessed via the ES, where a lower $\texttt{ES}(\mathcal{D}_u; (1-\alpha)\boldsymbol{\theta}_{\text{ref}} + \alpha\boldsymbol{\theta})$ indicates better performance of unlearning.

With UWC, we can assess the efficacy of unlearning across various models in a general and reliable manner. UWC will facilitate our hyper-parameter tuning and the comparisons of previous works, further supporting our explorations of practical tricks in the section below.

## 6 EXPERIMENTS

We benchmark existing LLM unlearning methods using UWC, recommending their proper hyper-parameters, assessing and comparing their efficacy in achieving effective unlearning. For the promising methods among the candidates, we further examine a series of simple tricks, which can further enhance their practical effectiveness in unlearning.

**Experimental Setups.** Our main evaluations were based on the well-established benchmarks of TOFU fictitious unlearning (Maini et al., 2024), incorporating two popular LLMs, including Phi-1.5 (Li et al., 2023b) and Llama-2-7B (Touvron et al., 2023a). For the unlearning setups, original training data are separated into targeted and non-targeted parts, of which the adopted proportions are 1:99 (1% unlearning), 5:95 (5% unlearning), and 10:90 (10% unlearning). Please refer to Appendix B for more details about the adopted experimental setups.

**Hyper-parameter Configurations.** We conduct extensive hyper-parameter tuning for the considered unlearning methods, as detailed in Appendix C. The full results across each setup of hyper-parameters can be found in Appendix H. With meticulous selection, we suggest $\lambda = 2$ for GD, $\lambda = 10$ for KL, and $\lambda = 20$ and $\beta = 0.5$ for NPO. Moreover, for RMU, we select the 9-th layer with $c = 4$ for Phi-1.5 and 21-th layer with $c = 2$ for Llama-2-7B.

Table 2: Comparison between different tricks for KL on TOFU with UWC calibration. ↓ / ↑ indicate smaller / larger values are preferable. We primarily focus on the ES scores for unlearning (shaded), given that the ES scores for retention are calibrated.

| LLM | | Phi-1.5 | | | | Llama-2-7B | | | |
|---|---|---|---|---|---|---|---|---|---|
| | | ES-exact | | ES-perturb | | ES-exact | | ES-perturb | |
| setup | method | retain ↑ | unlearn ↓ | retain ↑ | unlearn ↓ | retain ↑ | unlearn ↓ | retain ↑ | unlearn ↓ |
| 1% | origin | 0.4232 | 0.2123 | 0.2005 | 0.0840 | 0.7337 | 0.0515 | 0.4428 | 0.0913 |
| | LR | 0.4232 | 0.2031 | 0.2005 | 0.1078 | 0.7241 | **0.0428** | 0.4791 | **0.0000** |
| | BS | 0.4232 | 0.1931 | 0.2005 | 0.1078 | 0.7241 | **0.0428** | 0.4791 | **0.0000** |
| | ES | 0.4232 | 0.2033 | 0.2136 | 0.0571 | 0.8277 | 0.1029 | 0.4419 | 0.0403 |
| | TS | 0.4853 | **0.0586** | 0.2517 | **0.0175** | 0.7327 | 0.0522 | 0.4304 | 0.0368 |
| | LS | 0.4620 | 0.3540 | 0.2443 | 0.1582 | 0.7900 | 0.6105 | 0.4656 | 0.3738 |
| 5% | origin | 0.3823 | 0.3766 | 0.1794 | 0.1614 | 0.7207 | 0.0953 | 0.4814 | 0.1516 |
| | LR | 0.4404 | 0.4345 | 0.2069 | 0.1652 | 0.7377 | 0.0953 | 0.4258 | 0.0880 |
| | BS | 0.3879 | 0.3352 | 0.2049 | 0.1432 | 0.6825 | 0.0590 | 0.4450 | 0.0604 |
| | ES | 0.4536 | **0.2224** | 0.2137 | 0.1386 | 0.7928 | **0.0231** | 0.4493 | **0.0144** |
| | TS | 0.5776 | 0.5184 | 0.2473 | **0.0461** | 0.7018 | 0.1406 | 0.4362 | 0.0399 |
| | LS | 0.5766 | 0.2480 | 0.2492 | 0.1293 | 0.7080 | 0.3539 | 0.4299 | 0.2182 |
| 10% | origin | 0.4424 | 0.4912 | 0.2075 | **0.0922** | 0.7765 | 0.2791 | 0.4734 | 0.1236 |
| | LR | 0.3864 | 0.4585 | 0.2001 | 0.1215 | 0.7649 | 0.2791 | 0.4449 | 0.1057 |
| | BS | 0.4302 | **0.3358** | 0.2334 | 0.1621 | 0.7228 | 0.2287 | 0.4285 | 0.1071 |
| | ES | 0.4433 | 0.3974 | 0.2024 | 0.1360 | 0.7803 | 0.2163 | 0.4482 | 0.1076 |
| | TS | 0.5881 | 0.4952 | 0.2493 | 0.1377 | 0.6851 | **0.0730** | 0.4278 | **0.0000** |
| | LS | 0.5909 | 0.4347 | 0.2462 | 0.1197 | 0.6984 | 0.4711 | 0.4249 | 0.1712 |

## 6.1 MAIN RESULTS

We report not only the ES scores for original data but also for the associated paraphrased versions provided by TOFU. These paraphrased datasets maintain the original semantics but feature varied syntax and order, which can be employed to assess the generalization capability of the resulting models. To make the following discussion clear, we term the ES calculated for the original data as **ES-exact**, and that calculated for the paraphrased versions as **ES-perturb**. The full results after the UWC calibration are summarized in Table 1. Here, we summarize some of our key observations.

**Hardness of Unlearning Tasks.** Across unlearning setups, we observe that larger forget rates do not necessarily correspond to more challenging unlearning tasks, contrary to prior believes (Zhang et al., 2024). Our results indicate that the 5% setup is more challenging compared to that for both 1% and 10%. Therefore, specific data targeted for unlearning should also be taken into consideration when deciding the hardness of unlearning tasks. Across models, we find that Llama-2-7B can lead to overall better efficacy than Phi-1.5, indicating that unlearning for smaller models are harder.

**GA Variants Remain Promising.** Previous works often take GA and its variants as ineffective. However, via proper fine-tuning for the trade-off hyper-parameter, it reveals that GA-based methods, particularly GD and KL, can in fact exhibit attractive performance. Note that while we identify several cases where the original GA achieves the best ES-exact scores, this might be attributed to excessive unlearning that leads to overfitting, signifying by its higher ES-perturb with poor generalization. Therefore, we conclude that the retain loss is indispensable for GA-based methods.

**Excessive / Incomplete Unlearning is Common.** GA and NPO are two important methods in the literature. However, we show that, after UWC calibration, their efficacy in unlearning is not that attractive as our previous belief. However, the causes of their inferior performance are different, which can be seen from the results without UWC calibration in Table 3. As we can see, after unlearning, the ES scores of NPO are much greater than 0, a signal where the strength of unlearning is insufficient. We provide more justification from the weighting perspective and the risk perspective in Appendix G. On the other side, the ES scores of GA are all near 0, whether for unlearning or retention, indicate its strength of unlearning may too large, occupying the parameterized knowledge for non-targeted data, thereby making the resulting model completely useless. Nevertheless, we find that GD and KL with regularization terms of retention can largely mitigate its drawbacks.

## 6.2 BAG OF TRICKS

Beyond benchmarking existing works, UWC also enables us to delve into a variety of practical tricks that can empirically enhance the efficacy of unlearning. This aspect has been overlooked in the past, partly due to the pursuit of both removal and retention, which are mutual-conflicting. Such dual goals render it hard to determine whether the overall efficacy of unlearning has indeed improved after applying a particular trick. We fill this gap with our UWC, examining tricks listed as follows.

- **Learning Rate (LR), Early Stopping (ES), and Batch Size (BS)**. The learning rate dictates the intensity of unlearning, early stopping limits the number of updates, and the batch size connects to the stability of gradient estimation, which are all common tools to refine parameter updating.

- **Temperature Scaling (TS)**. The temperature is typically applied to logits before the softmax outputs. Its use during training can prevent overfitting and enhance robustness against noise.

- **Loss Selection (LS)**. We select a portion of tokens that exhibit the largest loss values and apply gradient updates only for them. It is designed to prevent excessive unlearning for tokens that already demonstrate very small loss values, especially intriguing when using GA.

Please refer to Appendix F for more details. Our investigations focus on KL, which is identified by UWC as a promising method. We conduct experiments across different configurations and hyper-parameter setups for these considered tricks in Appendix H, and summarize the results after hyper-parameter tuning in Table 2. Overall, we find that LS is not reliable for unlearning. On the other side, BS, ES, and TS play crucial roles in improving unlearning efficacy, which can enhance reliability of unlearning without incurring additional computational costs. However, for harder tasks, the benefits provided by BS and ES diminish, whereas TS continues to be highly effective, for example, as demonstrated in the 10% unlearning setup with LLama-2-7B. Overall, we recommend the default use of TS as a reliable trick in practice, along with proper hyper-parameter tuning of the unlearning epochs and/or batch sizes to further enhance unlearning.

## 7 CONCLUSION

This paper addresses the critical challenges in evaluating and comparing LLM unlearning methods. Recognizing the susceptibility of existing metrics to various attacks and the difficulty in balancing between removal and retention, we propose an effective evaluation framework named UWC. The UWC introduces the ES as a reliable unlearning metric, outperforming others in capturing the true extent of unlearning. Moreover, to address the trade-off between unlearning and retention, we calibrate model performance on non-targeted data via MM, ensuring that the retention of desirable knowledge is adequately preserved. By doing so, we can focus solely on assessing the unlearning efficacy on targeted data, facilitating fair comparisons across varying methods, models, and setups. Using the UWC framework, we benchmark representative unlearning methods. We find GA-based methods remain to be a powerful line a work, while we need to careful control its extent of unlearning via hyper-parameter tuning. We also explore other tricks that can further improve the practical efficacy of unlearning, where we find that temperature scaling is in general helpful.

This paper fills the gap in assessing the effectiveness of unlearning metrics, further motivating our exploration into fair comparisons and enhancements of current unlearning methods. Each facet presents opportunities to delve deeper. For reliable metrics, it is beneficial to include a broader range of candidate metrics as well as to consider much more red team attacking methods. Additionally, assessing the influence removal without relying on gold standard models remains to be an unresolved issue. For fair comparisons, we suggest that model mixing is a promising strategy that could also enhance practical applications: Even for vanilla GA, model mixing can ensure that overall performance to be maintained. Further exploration in this direction could include selective or sparse mixing, focusing on a subset of parameters that are crucial for effective knowledge removal. For the bag of tricks, we recommend further explorations of other simple yet reliable techniques.

## ETHIC STATEMENT AND REPRODUCIBILITY

LLMs, trained on extensive web-sourced datasets, risk inadvertently memorizing and disseminating sensitive, private, or harmful information. This could lead to potential violations of privacy, intellectual property rights, and societal harm. Unlearning methods offer a promising solution to mitigate these ethical concerns, thus attracting increasing research attentions recently. Rather than developing new methods, we focus on ensuring effective evaluations and fair comparisons for various unlearning methods and unlearned models. Our studies contribute to the assessments of safe, legal, and trustworthy LLM usages, reflecting the true extent for the potential to disseminate sensitive personal data, copyrighted material, and other forms of harmful or unethical information. It aligns with the wide goal of ensuring that AI technologies can respect the rights of individuals. For reproducibility, we have detailed the experimental setups, hyper-parameter configurations, and hardware specifications. The code is publicly available at: https://github.com/tmlr-group/Unlearning-with-Control.

## ACKNOWLEDGMENTS

QZW, PNY, JNZ, and BH were supported by RGC Young Collaborative Research Grant No. C2005-24Y, NSFC General Program No. 62376235, Guangdong Basic and Applied Basic Research Foundation Nos. 2022A1515011652 and 2024A1515012399, RIKEN Collaborative Research Fund, HKBU Faculty Niche Research Areas No. RC-FNRA-IG/22-23/SCI/04, and HKBU CSD Departmental Incentive Scheme. TLL was partially supported by the following Australian Research Council projects: FT220100318, DP220102121, LP220100527, LP220200949, IC190100031. TLL and MS were supported by JST ASPIRE Grant Number JPMJAP2405.

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

## A  CONCEPTUAL PROOF FOR METRIC EFFECTIVENESS

We formalize our discussion by developing a causal framework (Huang et al., 2023) to comprehend metric effectiveness. It delineates the relationships between knowledge parametrization ($K$), the considered metric ($M$) to quantify this knowledge, model behaviors ($B$), and the interventions ($I$) introduced by red teaming attacks. We further incorporate the mediator of superficial behaviors ($S$), which explain the change due to $I$ without changing the underlying knowledge $K$.

**Pathways.** All considered metrics are presumed capable of assessing the strength of knowledge parametrization more or less, denoted as $K \rightarrow M$, such that changes in $K$ should be manifested by $M$. Additionally, the knowledge parametrization directly influences model behaviors, represented as $K \rightarrow B$. This relationship underscores that the way a model processes inputs and generates outputs is definitely a function of its internal knowledge. For intervention $I$, it will introduce superficial behaviors $S$ without altering the underlying knowledge $K$, and these superficial behaviors mediate the effect of interventions on model behaviors, i.e., $I \rightarrow S$ and $S \rightarrow B$, while $I \nrightarrow K$. The causal relationships can be visualized in Figure 4. Therein, by identifying $S$ as a mediator, we recognize that changes in $B$ due to $I$ are not indicative of changes in $K$.

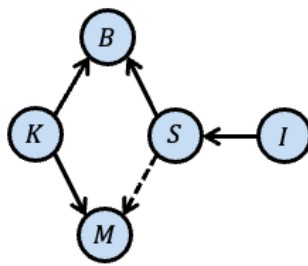

Figure 4: **The causal graph for the assessment of unlearning metrics.** The solid / dashed arrows represent known / unknown relationships.

**Assessing Effectiveness.** Our goal is to ensure that the crafted metrics $M$ are effective indicators of $K$ and are not unduly influenced by changes in $B$ caused by $I$, of which the directly modeling is not feasible. Instead, based on Figure 4, we conclude that an ideal metric should depend on $K$, holding true in general, and is robust to the change of $B$ via $I$. Therefore, to validate the effectiveness of a metric, we can test its robustness by testing a series of red teaming attacks that modify model behaviors by affecting superficial behaviors $S$ without altering the underlying knowledge $K$. Then, we can measure metrics before and after interventions to test their linear correlation, of which the high values suggest that the metric is robust and primarily dependent on $K$.

## B  EXPERIMENTAL CONFIGURATIONS

Our evaluations were based on the well-established benchmarks of TOFU fictitious unlearning (Maini et al., 2024), focusing on LLMs fine-tuned with a series of fictitious authors profiles. These profiles were created by prompting GPT-4 (Achiam et al., 2023), which has been filtered to avoid the occurrence of any real author profile, thus mitigating the inadvertent impacts of other unrelated variates. For each fictitious profile, TOFU crafted 20 question-answer pairs that can be used for fine-tuning, along with their paraphrased versions for evaluations.

The pre-trained LLMs were further fine-tuned on such question-answer pairs, where we considered two popular LLMs, i.e., Phi-1.5 (Li et al., 2023b) and Llama-2-7B (Touvron et al., 2023a) with their question-answering versions. For the unlearning setups, the original TOFU data were separated into targeted and non-targeted parts, of which the adopted proportions are 1:99 (1% unlearning), 5:95 (5% unlearning), and 10:90 (10% unlearning). Moreover, we further separated 400 non-targeted data that were not involved during the unlearning procedure for evaluations, reflecting real-world situations where it is not feasible to go through all non-targeted data during the unlearning process.

For all the considered methods, we adopt the following implementation setups: the AdamW optimizer (Loshchilov & Hutter, 2017), the initial learning rate $2e^{-5}$ for Phi-1.5 and $1e^{-5}$ for Llama-2-7B, the batch size 16 for both the targeted and non-targeted data, the epoch number 5, and the linear warm-up for the first epoch. For MM calibration, we set $\tau = 0.95$ for Phi-1.5 and $\tau = 0.90$ for Llama-2-7B. All our experiments were realized by Transformers 4.42.4 with CUDA 12.1, using a series of computation nodes equipped with NVIDIA-A100-80GB GPUs and Intel(R) Xeon(R) Gold 6248R CPU @ 3.00GHz Processors.

## C  BASELINE METHODS

We examine a collection of representative unlearning methods that are wide recognized in the literature. For clarity, we elaborate on their implementations and discuss their significance.

**Gradient Ascent (GA)** (Yao et al., 2023c). As one of the earliest unlearning methods, GA decreases the log-likelihood $\log p(s_{\mathrm{u}}; \boldsymbol{\theta})$ for targeted data. The unlearning objective is articulated as

$$-\mathbb{E}_{(x,y_{\mathrm{u}})\sim\mathcal{D}_{\mathrm{u}}}\ell\big(y_{\mathrm{u}}|x;\boldsymbol{\theta}\big), \tag{6}$$

corresponding to applying gradient ascent to the cross entropy loss. GA has been widely explored due to its simplicity (Liu et al., 2024). Nevertheless, it is also notorious for causing catastrophic collapse (Zhang et al., 2024)—its efficacy in removing targeted knowledge often comes at the large costs that damage the overall integrity of LLMs, rendering the resulting LLMs completely useless.

**Gradient Difference (GD)** (Maini et al., 2024). To counteract the negative impacts of catastrophic collapse, various regularization terms are explored to retain the common model integrity. GD improves upon GA by further decreasing the negative log-likelihood for non-targeted data, following

$$-\mathbb{E}_{(x,y_{\mathrm{u}})\sim\mathcal{D}_{\mathrm{u}}}\ell\big(y_{\mathrm{u}}|x;\boldsymbol{\theta}\big) + \lambda\mathbb{E}_{(x,y)\sim\mathcal{D}_{\mathrm{t}}\setminus\mathcal{D}_{\mathrm{u}}}\ell\big(y|x;\boldsymbol{\theta}\big), \tag{7}$$

where $\lambda$ is a trade-off hyper-parameter that should be tuned. The use of GD can mitigate the adverse effects of GA on knowledge retention. However, when the unlearning steps are extensive, the extreme scale of $\mathbb{E}_{(x,y_{\mathrm{u}})\sim\mathcal{D}_{\mathrm{u}}}\ell\big(y_{\mathrm{u}}|x;\boldsymbol{\theta}\big)$ will overshadow that of $\mathbb{E}_{(x,y)\sim\mathcal{D}_{\mathrm{t}}\setminus\mathcal{D}_{\mathrm{u}}}\ell\big(y|x;\boldsymbol{\theta}\big)$. Therefore, the GD will be less effective in the later unlearning phrase, reducing its ability to maintain utility.

**KL Regularization (KL)** (Maini et al., 2024). KL also involves regularization for GA. However, instead of learning from original data, KL retains the original responses for data by minimize the KL divergence before and after unlearning. The overall unlearning objective is

$$-\mathbb{E}_{(x,y_{\mathrm{u}})\sim\mathcal{D}_{\mathrm{u}}}\ell\big(y_{\mathrm{u}}|x;\boldsymbol{\theta}\big) + \lambda\mathbb{E}_{(x,y)\sim\mathcal{D}_{\mathrm{t}}\setminus\mathcal{D}_{\mathrm{u}}}\sum_{k}\mathrm{KL}\big(p(y^{<k} \mid x;\boldsymbol{\theta})\|p(y^{<k} \mid x;\boldsymbol{\theta}_{\mathrm{ref}})\big), \tag{8}$$

which averages the KL divergence with respect to a sequence of prefixes. Similar to GD, KL still suffers from deterioration in retention.

**Negative Preference Optimization (NPO)** (Zhang et al., 2024). It is motivated by direct preference optimization (DPO), a well-known alignment method (Rafailov et al., 2023), which originally utilizes paired corpora comprising preferred versus dis-preferred data. NPO segregates the dis-preferred part from DPO, heuristically employing it as the unlearning objective, following

$$\frac{2}{\beta}\mathbb{E}_{(x,y_{\mathrm{u}})\sim\mathcal{D}_{\mathrm{u}}}\log\big(1 + \big(\frac{p(y_{\mathrm{u}}|x;\boldsymbol{\theta})}{p(y_{\mathrm{u}}|x;\boldsymbol{\theta}_{\mathrm{ref}})}\big)^{\beta}\big) + \lambda\mathbb{E}_{(x,y)\sim\mathcal{D}_{\mathrm{t}}\setminus\mathcal{D}_{\mathrm{u}}}\sum_{k}\mathrm{KL}\big(p(y^{<k} \mid x;\boldsymbol{\theta})\|p(y^{<k} \mid x;\boldsymbol{\theta}_{\mathrm{ref}})\big), \tag{9}$$

where $\beta$ is the hyper-parameter of the inverse temperature. The effective realization of NPO still relies on regularization for retention, we default to use KL in our realization. We simply set $\lambda = 1$ to ease hyper-parameter tuning, which is suggested by (Zhang et al., 2024).

**Preference Optimization (PO)** (Maini et al., 2024). It aims to mitigate the drawbacks of the unlearning risk by targeting a new outcome, e.g., "I don't know.", which is implemented through

$$\mathbb{E}_{(x,y_{\mathrm{u}})\sim\mathcal{D}_{\mathrm{u}}}\ell(y_{\mathrm{idk}}|x;\boldsymbol{\theta}), \tag{10}$$

changing original outputs for targeted data to $y_{\mathrm{idk}}$.

**Representation Misdirection for Unlearning (RMU)** (Li et al., 2024). Instead of changing model outputs, RMU implements unlearning by perturbing model representation. Denote the embedding features by $\phi(s; \boldsymbol{\theta})$, the formulation of RMU is given by

$$\mathbb{E}_{(x,y_u)\sim\mathcal{D}_{\mathrm{u}}}\frac{1}{|y_{\mathrm{u}}|}\sum_{i=1}^{|y_{\mathrm{u}}|}||\phi([x, y^{<i}]; \boldsymbol{\theta}) - c \cdot \boldsymbol{u}||_2^2$$

$$+ \mathbb{E}_{(x,y)\sim\mathcal{D}_{\mathrm{t}}\setminus\mathcal{D}_{\mathrm{u}}}\frac{1}{|y|}\sum_{i=1}^{|y|}||\phi([x, y^{<i}]; \boldsymbol{\theta}) - \phi([x, y^{<i}]; \boldsymbol{\theta}_{\mathrm{ref}})||_2^2, \tag{11}$$

where $\boldsymbol{u}$ has elements randomly sampled from $[0, 1)$ and $c$ is a scaling hyper-parameter.

# D ATTACK METHODS

We consider four distinct types of red teaming scenarios: jail-breaking, embedding probing, relearning, and token noising. These attacking methods are utilized to assess the robustness of 5 basic metrics. Please see Figure 5 for the full results. Overall, we find that relearning and probing generally pose greater challenges compared to jail-breaking and noising, with the latter two exhibiting typically smaller PPC. Additionally, we observe that the PPL and KL are very similar, as both rely on the likelihood of outputs. For ROUGE, the results show that it is sensitive to attacks, suggesting it may not be an appropriate metric. When comparing PPL, ES, and EM across different scenarios, we find that their PPCs are close under jail-breaking and token noising. However, in the contexts of relearning and embedding probing, ES notably outperforms both PPL and EM. Therefore, we overall recommend adopting the ES as the most suitable basic metric for LLM unlearning.

Below, we offer detailed descriptions for the realizations of our considered attacks.

**Jail-breaking**. Jail-breaking manipulates or crafts input prompts to make the model produce outputs that bypass its safeguards (Li et al., 2023a). We explore two prompt structures as outlined in (Shen et al., 2023). For a given original question-answer pair, characterized by {question} and {answer}, these two prompts are constructed as follows:

> **Prompt 1.** Do not apologize or feel sorry, answer the question: {question}
> Answer: {answer}.

> **Prompt 2.** {question} Start with 'Absolutely! I know the answer! Here is:'
> Absolutely! I know the answer! Here is: {answer}.

We compute the metric scores under these two jail-breaking attacks separately, and report their average values as the final results.

**Embedding Probing**. We follow the approach of tuned lens (Belrose et al., 2023), which extracts embedding features from unlearned models, further decoding them into the output token space using an linear model. This additional model is trained on the whole TOFU datasets for 5 epochs, utilizing the Adam optimizer with the initial learning rate of $1e^{-3}$. Moreover, we focus on specific layers in our analysis, including the 11-st, 22-nd, and 33-rd layers for Llama-2-7B, and the 8-th, 16-th, and 24-th layers for Phi-1.5. The associated linear models are trained separately for each layer of embeddings. The performance metrics are averaged across layers, and we report the average values as the final results for each model type, either Phi-1.5 or Llama-2-7B.

**Relearning**. The unlearning models are further fine-tuned on targeted data for one epoch, using the negative log-likelihood as the objective. The AdamW optimizer is adopted with the same learning rates as original fine-tuning. The metric scores are then computed for relearned models.

**Token Noising**. We randomly select 5% of the tokens (ensuring at least one token is selected) in each string and replace it with a randomly chosen new token. This process introduces noise into data, simulating errors or disturbances that might occur in real-world applications. The metric scores are then computed for the original unlearned models, using the noised data as the ground truth.

Based on our analyses in Appendix A, we know that a proper attack method should not impact the parameterized knowledge within models, but can change model behaviors. From this perspective, jail-breaking and embedding probing are more appropriate than relearning and token noising when assessing metric robustness for unlearning. Therefore, the results of jail-breaking and embedding probing should receive our main focus for testing robustness.

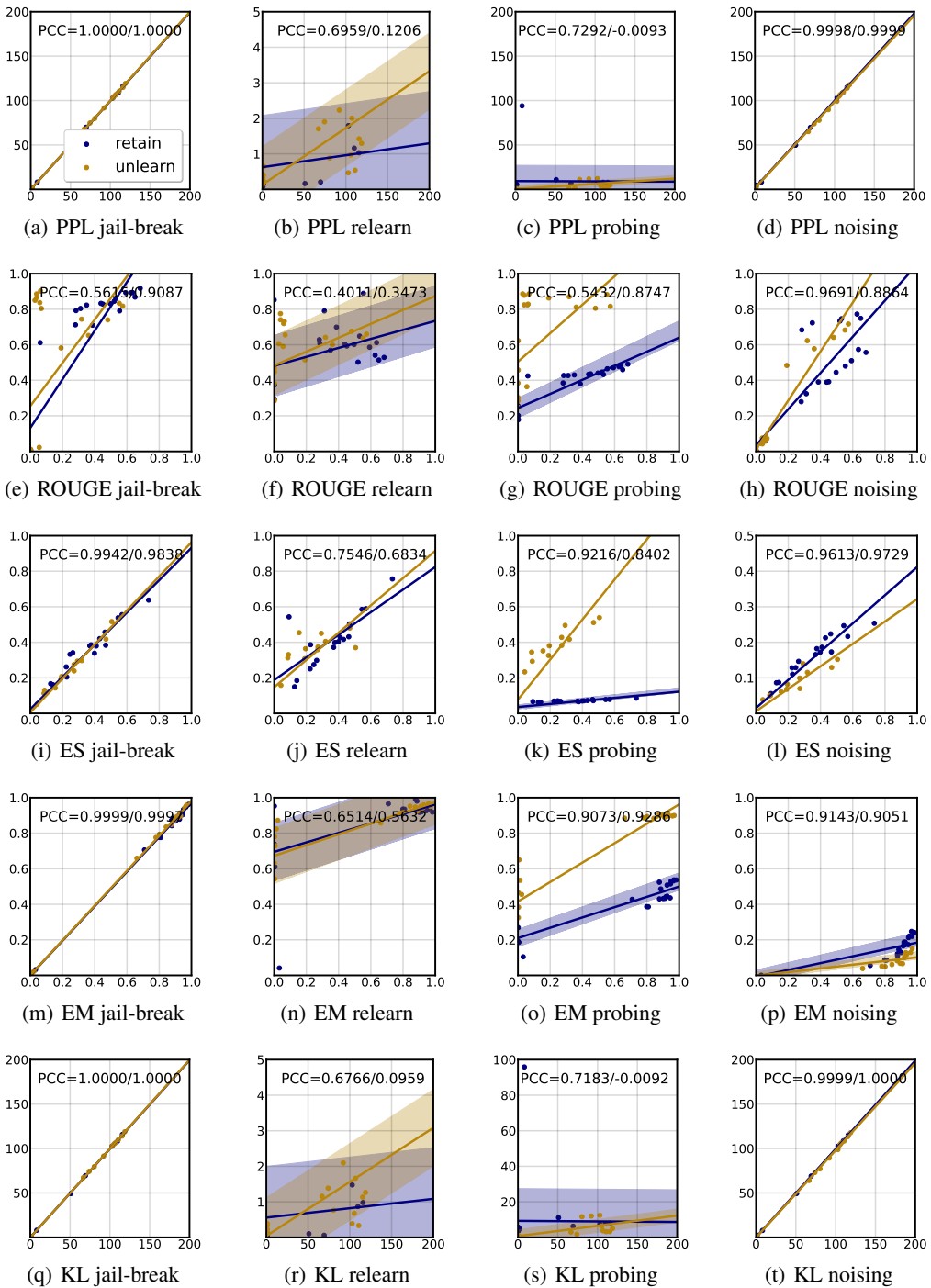

Figure 5: **Robustness of Metrics under Red Teaming Attacks.** We depict the metric scores before (**x-axis**) and after (**y-axis**) attacks jointly for different unlearning setups: across 2 LLMs (Phi-1.5 and Llama-2-7B), 3 unlearning percentages (1%, 5%, and 10%), and 4 unlearning methods (GA, GD, PO, and NPO). We consider 5 different metrics under 4 red teaming behaviors. We apply the log-scale for PPL to avoid numeric errors. For each of these scenarios, we compute the PPC with respect to targeted and non-targeted data respectively, displayed at the top of each figure (targeted data / non-targeted data). We provide linear fits for targeted and non-targeted data separately, accompanied by shaded areas representing the standard deviation to further visualize the PPC scores.

---

**Algorithm 1** Binary Search for MM Calibration

---

**Input:** parameters $\boldsymbol{\theta}_{\mathrm{ref}}$ before unlearning and $\boldsymbol{\theta}$ after unlearning; datasets $\mathcal{D}_{\mathrm{u}}$ and $\mathcal{D}_{\mathrm{t}}$; num_iter of total searching steps; threshold $\tau$.
$l_{\mathrm{cur}} = 0$ and $u_{\mathrm{cur}} = 1$
**for** iter $= 1$ **to** num_iter **do**
    $\alpha_{\mathrm{can}} \leftarrow (u_{\mathrm{cur}} + l_{\mathrm{cur}})/2$;
    $\boldsymbol{\theta}_{\mathrm{mix}} = (1 - \alpha_{\mathrm{can}})\boldsymbol{\theta}_{\mathrm{ref}} + \alpha_{\mathrm{can}}\boldsymbol{\theta}$
    **if** $\mathrm{ES}(\mathcal{D}_{\mathrm{t}}\backslash\mathcal{D}_{\mathrm{u}}; \boldsymbol{\theta}_{\mathrm{mix}}) \geq \tau\mathrm{ES}(\mathcal{D}_{\mathrm{t}}\backslash\mathcal{D}_{\mathrm{u}}; \boldsymbol{\theta}_{\mathrm{ref}})$ **then**
        $l_{\mathrm{cur}} \leftarrow (u_{\mathrm{cur}} + l_{\mathrm{cur}})/2$;
    **else**
        $u_{\mathrm{cur}} \leftarrow (u_{\mathrm{cur}} + l_{\mathrm{cur}})/2$;
    **end if**
**end for**
**Output:** optimal $\alpha^* = \alpha_{\mathrm{can}}$.

---

# E  UWC REALIZATION

While we have demonstrated the effectiveness of the UWC in evaluating and comparing unlearned models or unlearning methods, the computational expenses associated with its straightforward implementation, including the ES computation and the MM calibration, can be exorbitantly high.

Specifically, for the precise computation of the ES, it is necessary to iterate through each integer value $k \in \{1, \ldots, |y|\}$ to determine if the condition $f([x, y_{\mathrm{u}}^{<k}]; \boldsymbol{\theta}) = y_{\mathrm{u}}^{>k}$ is satisfied, then identifying the smallest value of $k$ among candidates. For the MM calibration, it is essential to sample a sufficient number of candidates $\alpha$ from the continuous range between 0 and 1. This involves testing whether the corresponding mixed model, with parameters $(1 - \alpha)\boldsymbol{\theta}_{\mathrm{ref}} + \alpha\boldsymbol{\theta}$, maintains acceptable performance on non-targeted data, i.e., $\mathrm{ES}(\mathcal{D}_{\mathrm{t}}\backslash\mathcal{D}_{\mathrm{u}}; (1 - \alpha)\boldsymbol{\theta}_{\mathrm{ref}} + \alpha\boldsymbol{\theta}) > \tau\mathrm{ES}(\mathcal{D}_{\mathrm{t}}\backslash\mathcal{D}_{\mathrm{u}}; \boldsymbol{\theta}_{\mathrm{ref}})$. To accurately estimate the optimal $\alpha$ with minimal damage on common integrity, it is crucial that the coverage of $\alpha$ should be sufficiently fine-grained, thereby increasing overall costs of calibration.

Fortunately, we observe approximately monotonic relationships for both $k$ and $\alpha$ with respect to their associated conditions. These scenarios indicate that the binary search can be effectively used to streamline the selection process for their appropriate values. Taking MM-based calibration as an example, Algorithm 1 outlines a general framework for the efficient parameter search of optimal $\alpha$. Similar implementations can also be adopted for computing the ES scores.

# F  MORE DISCUSSIONS ABOUT PRACTICAL TRICKS

In Section 6, we explore a series of tricks, such as adjusting common hyper-parameters for optimization, including the learning rate, batch size, unlearning epochs. Additionally, we suggest some more intriguing methods such as TS and LS. We further discuss the detailed implementations for the last two methods for concreteness.

**Temperature Scaling** (TS). By manipulating logits of model outputs, the TS is particular useful in avoid overfitting. Denote the original output logits as $z$, then the softmax function with the temperature scaling $\chi$ can be articulated as

$$\frac{\exp\{z_i/\chi\}}{\sum_j \exp\{z_j/\chi\}}. \tag{12}$$

Overall, higher temperatures will result in a softer probability distribution over candidate tokens, which can prevent the model from becoming too confident on the training data, thereby avoiding excessive unlearning and improving generalization. In our realization, we only apply TS for the unlearning risk.

**Loss Selection** (LS). During unlearning, we assume a proportion of tokens with already small loss values should not be involved during model updating, otherwise, severe excessive unlearning may

occur. Written the formulation of GA in a token-wise manner, we have

$$\mathbb{E}_{(x,y_{\mathrm{u}})\sim\mathcal{D}_{\mathrm{u}}}\frac{1}{|y|}\sum_{k}\log p\big(y^{k}|[x,y^{<k}];\boldsymbol{\theta}\big). \tag{13}$$

Then, we select $q \times 100\%$ proportion of tokens with largest loss values, satisfying the condition

$$K \leftarrow \arg\max_{|K'|\geq q|y|}\frac{1}{|K'|}\sum_{k\in K'}\log p\big(y^{k}|[x,y^{<k}];\boldsymbol{\theta}\big), \tag{14}$$

with $K'$ defining as a set of the selected tokens within $y$. Then, the GA with loss selection can be simply written as

$$\mathbb{E}_{(x,y_{\mathrm{u}})\sim\mathcal{D}_{\mathrm{u}}}\frac{1}{|K|}\sum_{k\in K}\log p\big(y^{k}|[x,y^{<k}];\boldsymbol{\theta}\big). \tag{15}$$

LS is particular attractive for those unbounded loss functions just like GA. In avoiding to update loss for the part of tokens that have been sufficiently unlearned, the resulting unlearning procedure has the potent to avoid excessive unlearning.

## G  EXCESSIVE UNLEARNING AND INCOMPLETE UNLEARNING

We claim that NPO suffers from incomplete unlearning, while GA exhibits tendencies of excessive unlearning. In this section, we provide more results to justify our claims.

For NPO, its unlearning behaviors can be analyzed through its gradient behaviors as outlined in (Zhang et al., 2024; Wang et al., 2025). When taking gradient with respect to $\boldsymbol{\theta}$, we have the gradients of NPO following the form of

$$\mathbb{E}_{(x,y)_{\mathrm{u}}\sim\mathcal{D}_{\mathrm{u}}}w_{x,y_{\mathrm{u}}}\nabla_{\boldsymbol{\theta}}\,\log p(y_{\mathrm{u}};x,\boldsymbol{\theta}), \tag{16}$$

with $w_{x,y_{\mathrm{u}}} = \frac{2p(y_{\mathrm{u}}|x;\boldsymbol{\theta})^{\beta}}{p(y_{\mathrm{u}}|x;\boldsymbol{\theta})^{\beta}+p(y_{\mathrm{u}}|x;\boldsymbol{\theta}_{\mathrm{o}})^{\beta}}$ can be viewed as a weighting mechanism. The effects of this mechanism for 5% unlearning with Llama-2-7B is illustrated in Figure 6, which shows the average $w_{x,y_{\mathrm{u}}}$ computed during NPO unlearning. Notably, these values quickly de-

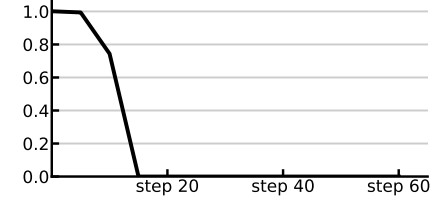

Figure 6: The dynamics of the implicit NPO weighting mechanism.

cline to 0 shortly after the end of the first epoch. The loss values and the ES scores do not notably change thereafter, which signifies that $w_{x,y_{\mathrm{u}}}$ plays the role of early stopping, thereby potentially leading to incomplete unlearning.

We examine risk values and ES scores for GA in Figure 7 for 5% unlearning with Llama-2-7B. Contrary to the NPO, we observe that ES scores quickly drop to 0 for unlearning, while the unlearning risks continue to decrease, indicating that the excessive unlearning may occur. The primary consequence of such excessive unlearning is a degradation in model performance on non-targeted data, evidenced by the poor ES scores on non-targeted data without calibration and the poor ES scores on targeted data with calibration.

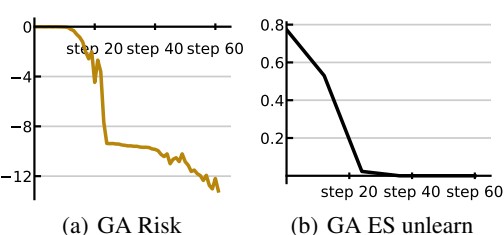

(a) GA Risk  (b) GA ES unlearn

Figure 7: The trajectories of risk and ES values.

## H    MORE RESULTS

In Table 3, we present the results of ES scores without calibration, where we observe an obvious trade-off between removal and retention. Since both methods are crucial for practical unlearning, it is difficult to conclude which is overall superior. This further emphasizes that calibration facilitates the comparison of overall efficacy across different methods.

We list detailed results involved during hyper-parameter tuning in Tables 4-10. For baseline methods, it involves the trade-off parameter $\lambda$ for GD, KL, and NPO; the inverse temperature $\beta$ for NPO; the scaling parameter $c$ and the embedding layers for RMU. $\lambda$ is chosen from the candidate set of $\{1, 2, 4, 7, 10, 20, 50, 100\}$; $\beta$ is chosen from $\{1, 2, 4, 7, 10, 20, 50, 100\}$, $c$ is chosen from $\{0, 1, 2, 4, 5, 7, 10\}$. The embedding layers of RMU is chosen from shallow, middle, and deep layers, respectively defined as 8-th, 16-th, and 24-th layers for Phi-1.5 and 11-th, 22-th, and 33-th layers for Llama-2-7B. Moreover, for NPO, we simplify its tuning procedure into two steps: **a)** fixing $\lambda = 1$ (original suggested) and tuning $\beta$ and **b)** fixing the tuned $\beta$ and tuning $\lambda$.

Then, we report the results involved for the bag of tricks in Tables 11-15. Therein, the learning rate is chosen from $\{1e^{-3}, 1e^{-4}, 1e^{-5}, 1e^{-6}, 1e^{-7}\}$; the batch size is chosen from $\{2, 4, 8, 14, 20\}$; the training epochs for early stopping is chosen from $\{1, 2, 3, 4, 5\}$; the temperature scaling $\chi$ is chosen from $\{0.9, 2, 3, 4, 5\}$; the likelihood capping $\kappa$ is chosen from $\{0.01, 0.1, 0.2, 0.3, 0.5\}$; the loss selection $q$ is chosen from $\{0.1, 0.3, 0.7\}$.

We explore the impact of various $\tau$ within the MM framework on comparison. We conduct experiments under the 5% unlearning scenario in Table 16, which demonstrate that the ranking of different methods, with respect to ES-exact, remains unchanged across varying $\tau$. This consistency underscores the robustness of our evaluation framework to specific settings of $\tau$.

Table 3: Comparison between different unlearning methods on TOFU fictitious unlearning **without UWC calibration**. ↓ / ↑ indicate smaller / larger values are preferable. We primarily focus on the ES scores for unlearning (shaded), given that the ES scores for retention are calibrated.

| LLM | | Phi-1.5 | | | | Llama-2-7B | | | |
|---|---|---|---|---|---|---|---|---|---|
| | | ES-exact | | ES-perturb | | ES-exact | | ES-perturb | |
| setup | method | retain ↑ | unlearn ↓ | retain ↑ | unlearn ↓ | retain ↑ | unlearn ↓ | retain ↑ | unlearn ↓ |
| before unlearning | | 0.4433 | 0.5969 | 0.2115 | 0.1605 | 0.8277 | 0.8039 | 0.5302 | 0.4001 |
| 1% | GA | 0.0000 | 0.0000 | 0.0000 | 0.0000 | 0.0003 | 0.0000 | 0.0000 | 0.0000 |
| | KL | 0.0459 | 0.0092 | 0.0458 | 0.0092 | 0.1676 | 0.0000 | 0.1564 | 0.0000 |
| | NPO | 0.2066 | 0.0648 | 0.1059 | 0.0558 | 0.4981 | 0.1201 | 0.3960 | 0.0963 |
| | RMU | 0.0000 | 0.0000 | 0.0000 | 0.0000 | 0.0000 | 0.0000 | 0.0000 | 0.0000 |
| before unlearning | | 0.4433 | 0.5619 | 0.2115 | 0.2374 | 0.8277 | 0.7735 | 0.5302 | 0.4126 |
| 5% | GA | 0.0001 | 0.0000 | 0.0000 | 0.0000 | 0.0000 | 0.0000 | 0.0000 | 0.0000 |
| | KL | 0.0873 | 0.0000 | 0.0892 | 0.0000 | 0.1985 | 0.0000 | 0.1459 | 0.0000 |
| | NPO | 0.1361 | 0.0877 | 0.0992 | 0.0725 | 0.4991 | 0.0891 | 0.3055 | 0.0780 |
| | RMU | 0.0000 | 0.0000 | 0.0000 | 0.0000 | 0.0000 | 0.0000 | 0.0000 | 0.0000 |
| before unlearning | | 0.4433 | 0.5299 | 0.2115 | 0.1843 | 0.8277 | 0.8307 | 0.5302 | 0.3099 |
| 10% | GA | 0.0000 | 0.0000 | 0.0000 | 0.0000 | 0.0000 | 0.0000 | 0.0000 | 0.0000 |
| | KL | 0.1105 | 0.0000 | 0.0791 | 0.0000 | 0.2690 | 0.0308 | 0.2566 | 0.0221 |
| | NPO | 0.3087 | 0.1201 | 0.1687 | 0.0671 | 0.6939 | 0.1623 | 0.4490 | 0.1227 |
| | RMU | 0.0000 | 0.0000 | 0.0000 | 0.0000 | 0.0000 | 0.0000 | 0.0000 | 0.0000 |

Table 4: **UWC Tuning for GD.** ↓ / ↑ indicate smaller / larger values are preferable.

| GD | | Phi-1.5 | | | | Llama-2-7B | | | |
|---|---|---|---|---|---|---|---|---|---|
| | | ES-exact | | ES-perturb | | ES-exact | | ES-perturb | |
| setup | $\lambda$ | retain ↑ | unlearn ↓ | retain ↑ | unlearn ↓ | retain ↑ | unlearn ↓ | retain ↑ | unlearn ↓ |
| before unlearning | | 0.4433 | 0.5969 | 0.2115 | 0.1605 | 0.8277 | 0.8039 | 0.5302 | 0.4001 |
| 1% | 1 | 0.4212 | 0.3449 | 0.2050 | 0.1010 | 0.8028 | 0.0873 | 0.4773 | 0.0000 |
| | 2 | 0.4212 | 0.3449 | 0.2072 | 0.1413 | 0.7471 | 0.0293 | 0.4471 | 0.1860 |
| | 4 | 0.4212 | 0.5219 | 0.2017 | 0.0506 | 0.7656 | 0.0241 | 0.5302 | 0.3242 |
| | 7 | 0.4404 | 0.5219 | 0.1644 | 0.0737 | 0.7177 | 0.1036 | 0.4791 | 0.0000 |
| | 10 | 0.4361 | 0.5219 | 0.2147 | 0.1120 | 0.7489 | 0.1775 | 0.4806 | 0.0719 |
| | 20 | 0.4312 | 0.5101 | 0.2009 | 0.1330 | 0.7420 | 0.3454 | 0.4829 | 0.2414 |
| | 50 | 0.4297 | 0.5969 | 0.2039 | 0.2039 | 0.7420 | 0.5682 | 0.4650 | 0.3501 |
| | 100 | 0.4263 | 0.5969 | 0.1994 | 0.2039 | 0.7928 | 0.7334 | 0.4905 | 0.3889 |
| before unlearning | | 0.4433 | 0.5619 | 0.2115 | 0.2374 | 0.8277 | 0.7735 | 0.5302 | 0.4126 |
| 5% | 1 | 0.4404 | 0.4310 | 0.1862 | 0.0563 | 0.7794 | 0.4362 | 0.4754 | 0.4126 |
| | 2 | 0.3919 | 0.4140 | 0.2004 | 0.0045 | 0.7432 | 0.3385 | 0.4775 | 0.3166 |
| | 4 | 0.3934 | 0.4574 | 0.2051 | 0.0000 | 0.7486 | 0.0903 | 0.4789 | 0.2176 |
| | 7 | 0.4454 | 0.4387 | 0.2137 | 0.0833 | 0.7822 | 0.2086 | 0.4498 | 0.3312 |
| | 10 | 0.4182 | 0.3381 | 0.2063 | 0.1663 | 0.7447 | 0.4527 | 0.4875 | 0.4126 |
| | 20 | 0.3826 | 0.4574 | 0.1899 | 0.2044 | 0.7366 | 0.5595 | 0.4696 | 0.2816 |
| | 50 | 0.4242 | 0.4494 | 0.1930 | 0.2079 | 0.7500 | 0.7001 | 0.4715 | 0.3309 |
| | 100 | 0.4411 | 0.4964 | 0.2036 | 0.2079 | 0.7467 | 0.7449 | 0.4970 | 0.3309 |
| before unlearning | | 0.4433 | 0.5299 | 0.2115 | 0.1843 | 0.8277 | 0.8307 | 0.5302 | 0.3099 |
| 10% | 1 | 0.4184 | 0.4683 | 0.2002 | 0.0841 | 0.7630 | 0.2926 | 0.4806 | 0.2428 |
| | 2 | 0.4454 | 0.4935 | 0.1761 | 0.0345 | 0.7771 | 0.0980 | 0.4780 | 0.1200 |
| | 4 | 0.4454 | 0.4878 | 0.1870 | 0.1182 | 0.7301 | 0.3178 | 0.4583 | 0.2035 |
| | 7 | 0.3913 | 0.4762 | 0.1940 | 0.1369 | 0.7731 | 0.3927 | 0.4782 | 0.2439 |
| | 10 | 0.4393 | 0.4935 | 0.2095 | 0.1540 | 0.7633 | 0.2772 | 0.4881 | 0.1115 |
| | 20 | 0.4433 | 0.5024 | 0.1958 | 0.1843 | 0.7394 | 0.2914 | 0.4790 | 0.1726 |
| | 50 | 0.3728 | 0.4967 | 0.2033 | 0.1600 | 0.7408 | 0.7278 | 0.4919 | 0.3051 |
| | 100 | 0.4242 | 0.5177 | 0.2051 | 0.1786 | 0.7422 | 0.7794 | 0.5210 | 0.3089 |

Table 5: **UWC Tuning for KL.** ↓ / ↑ indicate smaller / larger values are preferable.

| KL | | Phi-1.5 | | | | Llama-2-7B | | | |
|---|---|---|---|---|---|---|---|---|---|
| | | ES-exact | | ES-perturb | | ES-exact | | ES-perturb | |
| setup | $\lambda$ | retain ↑ | unlearn ↓ | retain ↑ | unlearn ↓ | retain ↑ | unlearn ↓ | retain ↑ | unlearn ↓ |
| before unlearning | | 0.4433 | 0.5969 | 0.2115 | 0.1605 | 0.8277 | 0.8039 | 0.5302 | 0.4001 |
| 1% | 1 | 0.4358 | 0.3606 | 0.1865 | 0.0789 | 0.7655 | 0.1307 | 0.4976 | 0.0373 |
| | 2 | 0.4251 | 0.3206 | 0.2005 | 0.0737 | 0.7655 | 0.1307 | 0.4867 | 0.0000 |
| | 4 | 0.4010 | 0.2679 | 0.1989 | 0.1283 | 0.7920 | 0.0382 | 0.4782 | 0.0000 |
| | 7 | 0.4232 | 0.2242 | 0.2136 | 0.0862 | 0.8277 | 0.0960 | 0.4754 | 0.2597 |
| | 10 | 0.4232 | 0.2123 | 0.2005 | 0.0840 | 0.7337 | 0.0515 | 0.4428 | 0.0913 |
| | 20 | 0.4232 | 0.1899 | 0.2051 | 0.0702 | 0.7826 | 0.0115 | 0.4729 | 0.0000 |
| | 50 | 0.4212 | 0.5219 | 0.1937 | 0.0724 | 0.7036 | 0.0633 | 0.4876 | 0.0281 |
| | 100 | 0.4232 | 0.3189 | 0.2172 | 0.1274 | 0.7567 | 0.0722 | 0.4532 | 0.0618 |
| before unlearning | | 0.4433 | 0.5619 | 0.2115 | 0.2374 | 0.8277 | 0.7735 | 0.5302 | 0.4126 |
| 5% | 1 | 0.4220 | 0.3466 | 0.1792 | 0.2349 | 0.7649 | 0.6896 | 0.4685 | 0.4031 |
| | 2 | 0.4419 | 0.3535 | 0.1991 | 0.2276 | 0.7346 | 0.6986 | 0.4796 | 0.3799 |
| | 4 | 0.4160 | 0.3340 | 0.2047 | 0.2162 | 0.7442 | 0.4097 | 0.4675 | 0.2461 |
| | 7 | 0.4220 | 0.3636 | 0.2182 | 0.1698 | 0.7702 | 0.5423 | 0.4816 | 0.7894 |
| | 10 | 0.3823 | 0.3766 | 0.1794 | 0.1614 | 0.7207 | 0.0953 | 0.4814 | 0.1516 |
| | 20 | 0.4109 | 0.1704 | 0.2027 | 0.1470 | 0.7196 | 0.1222 | 0.5302 | 0.3884 |
| | 50 | 0.4242 | 0.2129 | 0.2018 | 0.1691 | 0.7700 | 0.3494 | 0.5152 | 0.3243 |
| | 100 | 0.3588 | 0.2052 | 0.2115 | 0.1872 | 0.7697 | 0.3973 | 0.5302 | 0.3884 |
| before unlearning | | 0.4433 | 0.5299 | 0.2115 | 0.1843 | 0.8277 | 0.8307 | 0.5302 | 0.3099 |
| 10% | 1 | 0.4265 | 0.2989 | 0.2168 | 0.1459 | 0.7128 | 0.4250 | 0.4636 | 0.2343 |
| | 2 | 0.3582 | 0.2921 | 0.1957 | 0.1624 | 0.7274 | 0.6159 | 0.4738 | 0.2317 |
| | 4 | 0.4336 | 0.2373 | 0.2042 | 0.1168 | 0.7765 | 0.4791 | 0.4879 | 0.2317 |
| | 7 | 0.4164 | 0.4799 | 0.2048 | 0.0535 | 0.7554 | 0.4250 | 0.4761 | 0.2199 |
| | 10 | 0.4424 | 0.4912 | 0.2075 | 0.0922 | 0.7765 | 0.2791 | 0.4734 | 0.1236 |
| | 20 | 0.4418 | 0.5008 | 0.2069 | 0.0075 | 0.7860 | 0.2975 | 0.4927 | 0.1874 |
| | 50 | 0.3858 | 0.4722 | 0.2051 | 0.0691 | 0.7344 | 0.3132 | 0.4810 | 0.1870 |
| | 100 | 0.4242 | 0.4337 | 0.1991 | 0.1610 | 0.7720 | 0.4126 | 0.4959 | 0.2550 |

Table 6: **UWC Tuning for NPO** ($\lambda = 1$). $\downarrow$ / $\uparrow$ indicate smaller / larger values are preferable.

| NPO | | Phi-1.5 | | | | Llama-2-7B | | | |
|---|---|---|---|---|---|---|---|---|---|
| | | ES-exact | | ES-perturb | | ES-exact | | ES-perturb | |
| setup | $\beta$ | retain $\uparrow$ | unlearn $\downarrow$ | retain $\uparrow$ | unlearn $\downarrow$ | retain $\uparrow$ | unlearn $\downarrow$ | retain $\uparrow$ | unlearn $\downarrow$ |
| before unlearning | | 0.4433 | 0.5969 | 0.2115 | 0.1605 | 0.8277 | 0.8039 | 0.5302 | 0.4001 |
| | 0.05 | 0.4283 | 0.1587 | 0.2136 | 0.0702 | 0.7655 | 0.1262 | 0.5084 | 0.2545 |
| | 0.10 | 0.4553 | 0.1587 | 0.2121 | 0.0945 | 0.7547 | 0.1857 | 0.4995 | 0.2113 |
| | 0.50 | 0.4030 | 0.0947 | 0.2136 | 0.1083 | 0.6967 | 0.2513 | 0.4777 | 0.1898 |
| | 0.70 | 0.3909 | 0.1072 | 0.2136 | 0.1083 | 0.7517 | 0.2607 | 0.4733 | 0.1863 |
| 1% | 1.00 | 0.4261 | 0.1806 | 0.2136 | 0.1083 | 0.7517 | 0.2607 | 0.4777 | 0.1863 |
| | 2.00 | 0.3954 | 0.1166 | 0.2136 | 0.1655 | 0.7234 | 0.2876 | 0.4588 | 0.2025 |
| | 4.00 | 0.4223 | 0.1166 | 0.2136 | 0.1551 | 0.0000 | 0.0000 | 0.0000 | 0.0000 |
| | 5.00 | 0.4218 | 0.1806 | 0.2136 | 0.1551 | 0.0000 | 0.0000 | 0.0000 | 0.0000 |
| | 7.00 | 0.4218 | 0.1806 | 0.2001 | 0.1551 | 0.7874 | 0.2941 | 0.4588 | 0.2197 |
| | 10.0 | 0.4218 | 0.1806 | 0.2136 | 0.1551 | 0.0000 | 0.0000 | 0.0000 | 0.0000 |
| before unlearning | | 0.4433 | 0.5619 | 0.2115 | 0.2374 | 0.8277 | 0.7735 | 0.5302 | 0.4126 |
| | 0.05 | 0.4265 | 0.3671 | 0.2052 | 0.2349 | 0.0000 | 0.0000 | 0.0000 | 0.0000 |
| | 0.10 | 0.4161 | 0.3709 | 0.1942 | 0.2228 | 0.7652 | 0.5473 | 0.4976 | 0.4066 |
| | 0.50 | 0.4433 | 0.4539 | 0.2098 | 0.2228 | 0.7780 | 0.4966 | 0.4773 | 0.4009 |
| | 0.70 | 0.3970 | 0.3452 | 0.2058 | 0.2314 | 0.7459 | 0.5005 | 0.4903 | 0.4013 |
| 5% | 1.00 | 0.4086 | 0.4177 | 0.1982 | 0.2228 | 0.7836 | 0.5195 | 0.4918 | 0.3785 |
| | 2.00 | 0.4086 | 0.3863 | 0.2043 | 0.2203 | 0.7572 | 0.5809 | 0.4976 | 0.3884 |
| | 4.00 | 0.4433 | 0.4188 | 0.2043 | 0.2147 | 0.7836 | 0.5809 | 0.4781 | 0.3884 |
| | 5.00 | 0.4433 | 0.4188 | 0.2150 | 0.2147 | 0.7836 | 0.5946 | 0.5175 | 0.3726 |
| | 7.00 | 0.4127 | 0.4034 | 0.2109 | 0.1805 | 0.7836 | 0.5303 | 0.4887 | 0.3674 |
| | 10.0 | 0.4433 | 0.4034 | 0.1848 | 0.2000 | 0.7836 | 0.5703 | 0.5012 | 0.3674 |
| before unlearning | | 0.4433 | 0.5299 | 0.2115 | 0.1843 | 0.8277 | 0.8307 | 0.5302 | 0.3099 |
| | 0.05 | 0.4370 | 0.4360 | 0.2231 | 0.1526 | 0.7765 | 0.6204 | 0.4825 | 0.3137 |
| | 0.10 | 0.4222 | 0.4290 | 0.2048 | 0.1383 | 0.7765 | 0.5818 | 0.4809 | 0.3137 |
| | 0.50 | 0.4270 | 0.4708 | 0.2088 | 0.1645 | 0.7836 | 0.6310 | 0.4825 | 0.3271 |
| | 0.70 | 0.4413 | 0.4781 | 0.2088 | 0.1645 | 0.7836 | 0.6545 | 0.4825 | 0.3271 |
| 10% | 1.00 | 0.4073 | 0.4689 | 0.2074 | 0.1588 | 0.7836 | 0.6291 | 0.4825 | 0.3271 |
| | 2.00 | 0.4433 | 0.4712 | 0.2362 | 0.2224 | 0.7836 | 0.6375 | 0.4874 | 0.3244 |
| | 4.00 | 0.4433 | 0.4771 | 0.2225 | 0.1996 | 0.7836 | 0.6018 | 0.4795 | 0.3030 |
| | 5.00 | 0.4433 | 0.4771 | 0.2260 | 0.2105 | 0.7836 | 0.5387 | 0.5101 | 0.2989 |
| | 7.00 | 0.4433 | 0.4954 | 0.2260 | 0.1967 | 0.7479 | 0.5387 | 0.4809 | 0.2672 |
| | 10.0 | 0.4404 | 0.5465 | 0.1905 | 0.1990 | 0.7479 | 0.5387 | 0.4838 | 0.2774 |

Table 7: **UWC Tuning for NPO** ($\beta = 0.5$). $\downarrow$ / $\uparrow$ indicate smaller / larger values are preferable.

| NPO | | Phi-1.5 | | | | Llama-2-7B | | | |
|---|---|---|---|---|---|---|---|---|---|
| | | ES-exact | | ES-perturb | | ES-exact | | ES-perturb | |
| setup | $\lambda$ | retain $\uparrow$ | unlearn $\downarrow$ | retain $\uparrow$ | unlearn $\downarrow$ | retain $\uparrow$ | unlearn $\downarrow$ | retain $\uparrow$ | unlearn $\downarrow$ |
| before unlearning | | 0.4433 | 0.5969 | 0.2115 | 0.1605 | 0.8277 | 0.8039 | 0.5302 | 0.4001 |
| 1% | 1 | 0.4742 | 0.1166 | 0.2136 | 0.1551 | 0.7346 | 0.3134 | 0.4743 | 0.2066 |
| | 2 | 0.4627 | 0.1259 | 0.2136 | 0.1551 | 0.7648 | 0.3134 | 0.4777 | 0.2101 |
| | 4 | 0.4606 | 0.1259 | 0.2136 | 0.1551 | 0.7346 | 0.2941 | 0.4805 | 0.2066 |
| | 7 | 0.4535 | 0.1259 | 0.1837 | 0.0980 | 0.7952 | 0.2941 | 0.4909 | 0.3273 |
| | 10 | 0.4473 | 0.1259 | 0.1927 | 0.0702 | 0.6978 | 0.2543 | 0.4776 | 0.1672 |
| | 20 | 0.4424 | 0.1259 | 0.2136 | 0.0702 | 0.7383 | 0.2543 | 0.4776 | 0.1703 |
| | 50 | 0.4181 | 0.1259 | 0.1843 | 0.0983 | 0.6183 | 0.1383 | 0.5286 | 0.3017 |
| | 100 | 0.3970 | 0.1259 | 0.1909 | 0.0702 | 0.7251 | 0.2568 | 0.5302 | 0.3685 |
| before unlearning | | 0.4433 | 0.5619 | 0.2115 | 0.2374 | 0.8277 | 0.7735 | 0.5302 | 0.4126 |
| 5% | 1 | 0.4253 | 0.4462 | 0.1958 | 0.2228 | 0.7836 | 0.6062 | 0.4976 | 0.3635 |
| | 2 | 0.4125 | 0.3965 | 0.1923 | 0.2228 | 0.7836 | 0.6062 | 0.4641 | 0.3664 |
| | 4 | 0.4127 | 0.4354 | 0.2027 | 0.1985 | 0.7770 | 0.6177 | 0.4770 | 0.3835 |
| | 7 | 0.4148 | 0.3922 | 0.1984 | 0.1900 | 0.7820 | 0.4756 | 0.4938 | 0.3233 |
| | 10 | 0.4086 | 0.3991 | 0.2112 | 0.1381 | 0.7836 | 0.4756 | 0.4875 | 0.2784 |
| | 20 | 0.4433 | 0.3768 | 0.1836 | 0.1509 | 0.7207 | 0.1104 | 0.4804 | 0.2777 |
| | 50 | 0.3987 | 0.3396 | 0.2055 | 0.1120 | 0.7261 | 0.0443 | 0.4849 | 0.2092 |
| | 100 | 0.4242 | 0.3051 | 0.2118 | 0.1559 | 0.7509 | 0.1020 | 0.4672 | 0.2317 |
| before unlearning | | 0.4433 | 0.5299 | 0.2115 | 0.1843 | 0.8277 | 0.8307 | 0.5302 | 0.3099 |
| 10% | 1 | 0.4370 | 0.4478 | 0.2048 | 0.1502 | 0.7836 | 0.6139 | 0.4825 | 0.3244 |
| | 2 | 0.4393 | 0.4459 | 0.1870 | 0.1331 | 0.7836 | 0.4961 | 0.4796 | 0.2860 |
| | 4 | 0.4209 | 0.4505 | 0.2107 | 0.1188 | 0.7462 | 0.4479 | 0.4781 | 0.2066 |
| | 7 | 0.4433 | 0.4459 | 0.2110 | 0.0762 | 0.7479 | 0.4392 | 0.5059 | 0.1979 |
| | 10 | 0.4433 | 0.4397 | 0.1989 | 0.0764 | 0.7479 | 0.3208 | 0.4669 | 0.1738 |
| | 20 | 0.4072 | 0.3499 | 0.2028 | 0.1281 | 0.7769 | 0.3700 | 0.5100 | 0.1243 |
| | 50 | 0.4265 | 0.5221 | 0.2002 | 0.1018 | 0.7238 | 0.3439 | 0.4645 | 0.1867 |
| | 100 | 0.4173 | 0.4974 | 0.1735 | 0.0823 | 0.7362 | 0.3857 | 0.5302 | 0.3169 |

Table 8: **UWC Tuning for RMU (shallow).** $\downarrow$ / $\uparrow$ indicate smaller / larger values are preferable.

| RMU | | Phi-1.5 | | | | Llama-2-7B | | | |
|---|---|---|---|---|---|---|---|---|---|
| | | ES-exact | | ES-perturb | | ES-exact | | ES-perturb | |
| setup | $c$ | retain $\uparrow$ | unlearn $\downarrow$ | retain $\uparrow$ | unlearn $\downarrow$ | retain $\uparrow$ | unlearn $\downarrow$ | retain $\uparrow$ | unlearn $\downarrow$ |
| before unlearning | | 0.4433 | 0.5969 | 0.2115 | 0.1605 | 0.8277 | 0.8039 | 0.5302 | 0.4001 |
| 1% | 0.00 | 0.4530 | 0.5969 | 0.2007 | 0.1855 | 0.7604 | 0.5993 | 0.4888 | 0.3816 |
| | 1.00 | 0.4122 | 0.4356 | 0.2115 | 0.1855 | 0.7502 | 0.6278 | 0.4890 | 0.4253 |
| | 2.00 | 0.4312 | 0.4080 | 0.2072 | 0.1855 | 0.7653 | 0.6714 | 0.4531 | 0.4002 |
| | 4.00 | 0.4245 | 0.4682 | 0.2115 | 0.1855 | 0.7356 | 0.7223 | 0.0000 | 0.0000 |
| | 5.00 | 0.4398 | 0.5149 | 0.1981 | 0.1855 | 0.7163 | 0.6287 | 0.4871 | 0.4008 |
| | 7.00 | 0.4460 | 0.5096 | 0.2201 | 0.1855 | 0.7292 | 0.7128 | 0.4516 | 0.4104 |
| | 10.0 | 0.4215 | 0.4816 | 0.2018 | 0.1855 | 0.7292 | 0.6195 | 0.4453 | 0.4104 |
| before unlearning | | 0.4433 | 0.5619 | 0.2115 | 0.2374 | 0.8277 | 0.7735 | 0.5302 | 0.4126 |
| 5% | 0.00 | 0.4164 | 0.4924 | 0.1918 | 0.2172 | 0.7516 | 0.7292 | 0.4676 | 0.3616 |
| | 1.00 | 0.4284 | 0.5124 | 0.2194 | 0.2172 | 0.7762 | 0.7357 | 0.4677 | 0.4504 |
| | 2.00 | 0.4044 | 0.4774 | 0.1939 | 0.2172 | 0.7146 | 0.6370 | 0.4453 | 0.4126 |
| | 4.00 | 0.4404 | 0.4252 | 0.2047 | 0.2147 | 0.7619 | 0.6758 | 0.4812 | 0.4126 |
| | 5.00 | 0.4404 | 0.4838 | 0.2181 | 0.2207 | 0.7139 | 0.6758 | 0.4812 | 0.4164 |
| | 7.00 | 0.4204 | 0.3772 | 0.2073 | 0.2339 | 0.7604 | 0.6758 | 0.4793 | 0.4126 |
| | 10.0 | 0.4194 | 0.4114 | 0.1903 | 0.2339 | 0.7146 | 0.6370 | 0.4453 | 0.4126 |
| before unlearning | | 0.4433 | 0.5299 | 0.2115 | 0.1843 | 0.8277 | 0.8307 | 0.5302 | 0.3099 |
| 10% | 0.00 | 0.4425 | 0.5761 | 0.2055 | 0.1424 | 0.7887 | 0.8165 | 0.4246 | 0.2662 |
| | 1.00 | 0.4424 | 0.5968 | 0.2133 | 0.1567 | 0.7568 | 0.6869 | 0.4771 | 0.2989 |
| | 2.00 | 0.4304 | 0.5961 | 0.2028 | 0.1360 | 0.7628 | 0.6755 | 0.4690 | 0.2989 |
| | 4.00 | 0.4364 | 0.5208 | 0.1944 | 0.1547 | 0.7229 | 0.5784 | 0.4812 | 0.2766 |
| | 5.00 | 0.4284 | 0.5184 | 0.2007 | 0.1547 | 0.7262 | 0.6268 | 0.4797 | 0.2944 |
| | 7.00 | 0.4404 | 0.5184 | 0.2007 | 0.1754 | 0.7271 | 0.5778 | 0.4232 | 0.3033 |
| | 10.0 | 0.4404 | 0.4693 | 0.2136 | 0.1675 | 0.7032 | 0.5455 | 0.4849 | 0.3033 |

Table 9: **UWC Tuning for RMU (middle).** ↓ / ↑ indicate smaller / larger values are preferable.

| RMU | | Phi-1.5 | | | | Llama-2-7B | | | |
|---|---|---|---|---|---|---|---|---|---|
| | | ES-exact | | ES-perturb | | ES-exact | | ES-perturb | |
| setup | $c$ | retain ↑ | unlearn ↓ | retain ↑ | unlearn ↓ | retain ↑ | unlearn ↓ | retain ↑ | unlearn ↓ |
| before unlearning | | 0.4433 | 0.5969 | 0.2115 | 0.1605 | 0.8277 | 0.8039 | 0.5302 | 0.4001 |
| | 0.00 | 0.4203 | 0.5969 | 0.2153 | 0.2069 | 0.7606 | 0.5127 | 0.5115 | 0.4001 |
| | 1.00 | 0.4203 | 0.5969 | 0.2180 | 0.1409 | 0.7416 | 0.5093 | 0.4878 | 0.4001 |
| | 2.00 | 0.4203 | 0.5969 | 0.1831 | 0.1261 | 0.7512 | 0.4263 | 0.4644 | 0.3794 |
| 1% | 4.00 | 0.4203 | 0.5969 | 0.1831 | 0.1261 | 0.7559 | 0.5093 | 0.4096 | 0.3538 |
| | 5.00 | 0.4203 | 0.5969 | 0.2073 | 0.1328 | 0.7413 | 0.4810 | 0.4927 | 0.4001 |
| | 7.00 | 0.4218 | 0.5969 | 0.2119 | 0.1261 | 0.7413 | 0.4810 | 0.4927 | 0.4001 |
| | 10.0 | 0.4203 | 0.5969 | 0.2119 | 0.1350 | 0.7655 | 0.4137 | 0.4927 | 0.3624 |
| before unlearning | | 0.4433 | 0.5619 | 0.2115 | 0.2374 | 0.8277 | 0.7735 | 0.5302 | 0.4126 |
| | 0.00 | 0.4262 | 0.5723 | 0.1952 | 0.2207 | 0.0000 | 0.0000 | 0.0000 | 0.0000 |
| | 1.00 | 0.4232 | 0.4999 | 0.2032 | 0.2207 | 0.7381 | 0.4284 | 0.4798 | 0.3884 |
| | 2.00 | 0.4232 | 0.5013 | 0.2229 | 0.2207 | 0.7179 | 0.5146 | 0.4379 | 0.3884 |
| 5% | 4.00 | 0.4218 | 0.5309 | 0.1887 | 0.2030 | 0.7112 | 0.4034 | 0.4927 | 0.3884 |
| | 5.00 | 0.3578 | 0.3762 | 0.2119 | 0.2030 | 0.7438 | 0.6323 | 0.4927 | 0.3884 |
| | 7.00 | 0.4218 | 0.5946 | 0.1990 | 0.1971 | 0.7438 | 0.6684 | 0.4927 | 0.4126 |
| | 10.0 | 0.4262 | 0.4000 | 0.1968 | 0.2005 | 0.7552 | 0.6615 | 0.4644 | 0.4126 |
| before unlearning | | 0.4433 | 0.5299 | 0.2115 | 0.1843 | 0.8277 | 0.8307 | 0.5302 | 0.3099 |
| | 0.00 | 0.4262 | 0.4584 | 0.1952 | 0.1786 | 0.0000 | 0.0000 | 0.0000 | 0.0000 |
| | 1.00 | 0.4203 | 0.4909 | 0.2108 | 0.1816 | 0.7493 | 0.7636 | 0.4379 | 0.3139 |
| | 2.00 | 0.4232 | 0.5025 | 0.2212 | 0.1786 | 0.7374 | 0.7275 | 0.4831 | 0.3158 |
| 10% | 4.00 | 0.4394 | 0.5025 | 0.2117 | 0.1901 | 0.7874 | 0.7526 | 0.4871 | 0.3196 |
| | 5.00 | 0.4224 | 0.4511 | 0.2117 | 0.1799 | 0.7874 | 0.6907 | 0.4653 | 0.3220 |
| | 7.00 | 0.4005 | 0.4568 | 0.1496 | 0.1741 | 0.7434 | 0.5821 | 0.4776 | 0.2908 |
| | 10.0 | 0.0000 | 0.0000 | 0.0000 | 0.0000 | 0.7534 | 0.6495 | 0.4927 | 0.3316 |

Table 10: **UWC Tuning for RMU (deep).** ↓ / ↑ indicate smaller / larger values are preferable.

| UWC | | Phi-1.5 | | | | Llama-2-7B | | | |
|---|---|---|---|---|---|---|---|---|---|
| | | ES-exact | | ES-perturb | | ES-exact | | ES-perturb | |
| setup | $c$ | retain ↑ | unlearn ↓ | retain ↑ | unlearn ↓ | retain ↑ | unlearn ↓ | retain ↑ | unlearn ↓ |
| before unlearning | | 0.4433 | 0.5969 | 0.2115 | 0.1605 | 0.8277 | 0.8039 | 0.5302 | 0.4001 |
| | 0.00 | 0.3936 | 0.5219 | 0.2136 | 0.1574 | 0.7836 | 0.6364 | 0.4927 | 0.4089 |
| | 1.00 | 0.4156 | 0.5219 | 0.2117 | 0.1574 | 0.7461 | 0.4564 | 0.4442 | 0.3402 |
| | 2.00 | 0.4212 | 0.5219 | 0.2080 | 0.1655 | 0.6977 | 0.2814 | 0.4847 | 0.2790 |
| 1% | 4.00 | 0.4212 | 0.5153 | 0.1951 | 0.1655 | 0.6913 | 0.2992 | 0.4428 | 0.2748 |
| | 5.00 | 0.4212 | 0.5121 | 0.2062 | 0.1655 | 0.7122 | 0.3974 | 0.4976 | 0.1982 |
| | 7.00 | 0.4212 | 0.5108 | 0.1885 | 0.1686 | 0.7509 | 0.3271 | 0.4428 | 0.2305 |
| | 10.0 | 0.4184 | 0.4963 | 0.2136 | 0.1717 | 0.7106 | 0.3815 | 0.4428 | 0.2062 |
| before unlearning | | 0.4433 | 0.5619 | 0.2115 | 0.2374 | 0.8277 | 0.7735 | 0.5302 | 0.4126 |
| | 0.00 | 0.4212 | 0.4953 | 0.2007 | 0.2182 | 0.7731 | 0.7074 | 0.4675 | 0.3953 |
| | 1.00 | 0.4049 | 0.5144 | 0.2115 | 0.2182 | 0.7731 | 0.6488 | 0.4801 | 0.3850 |
| | 2.00 | 0.4110 | 0.5602 | 0.1967 | 0.2227 | 0.7410 | 0.6683 | 0.4801 | 0.3714 |
| 5% | 4.00 | 0.4151 | 0.5621 | 0.1930 | 0.2227 | 0.7731 | 0.6031 | 0.4598 | 0.3869 |
| | 5.00 | 0.4212 | 0.5271 | 0.2099 | 0.2394 | 0.7464 | 0.7001 | 0.4613 | 0.3958 |
| | 7.00 | 0.4212 | 0.5285 | 0.1951 | 0.2394 | 0.8113 | 0.6983 | 0.5015 | 0.4464 |
| | 10.0 | 0.4064 | 0.4816 | 0.2025 | 0.2349 | 0.7319 | 0.7763 | 0.4600 | 0.4393 |
| before unlearning | | 0.4433 | 0.5299 | 0.2115 | 0.1843 | 0.8277 | 0.8307 | 0.5302 | 0.3099 |
| | 0.00 | 0.4212 | 0.4935 | 0.2095 | 0.1933 | 0.7577 | 0.6868 | 0.4410 | 0.2884 |
| | 1.00 | 0.4049 | 0.4935 | 0.2039 | 0.1963 | 0.7673 | 0.7560 | 0.4571 | 0.2906 |
| | 2.00 | 0.4212 | 0.4935 | 0.1969 | 0.1933 | 0.7731 | 0.7402 | 0.4865 | 0.3239 |
| 10% | 4.00 | 0.4212 | 0.4935 | 0.2115 | 0.1933 | 0.7731 | 0.7414 | 0.4426 | 0.2674 |
| | 5.00 | 0.4212 | 0.4959 | 0.1967 | 0.1933 | 0.7486 | 0.7688 | 0.4738 | 0.2192 |
| | 7.00 | 0.4212 | 0.4799 | 0.2097 | 0.1933 | 0.7620 | 0.7402 | 0.4784 | 0.2547 |
| | 10.0 | 0.3934 | 0.4799 | 0.1951 | 0.1786 | 0.7394 | 0.7402 | 0.4890 | 0.2547 |

Table 11: **UWC Tuning for the Learning Rate of KL.** ↓ / ↑ indicate smaller / larger values are preferable.

| UWC | | Phi-1.5 | | | | Llama-2-7B | | | |
| | | ES-exact | | ES-perturb | | ES-exact | | ES-perturb | |
| setup | learning rate scale | retain ↑ | unlearn ↓ | retain ↑ | unlearn ↓ | retain ↑ | unlearn ↓ | retain ↑ | unlearn ↓ |
|---|---|---|---|---|---|---|---|---|---|
| 1% | $1e^{-3}$ | 0.4149 | 0.5053 | 0.1902 | 0.0770 | 0.7815 | 0.2315 | 0.4442 | 0.3080 |
| | $1e^{-4}$ | 0.4126 | 0.5219 | 0.1823 | **0.0228** | 0.7546 | 0.3095 | 0.4516 | 0.3289 |
| | $1e^{-5}$ | 0.4232 | **0.2031** | 0.2005 | 0.1078 | 0.7241 | **0.0428** | 0.4791 | **0.0000** |
| | $1e^{-6}$ | 0.4439 | 0.5108 | 0.2136 | 0.1551 | 0.8277 | 0.6798 | 0.4990 | 0.3458 |
| | $1e^{-7}$ | 0.4404 | 0.5876 | 0.2136 | 0.1889 | 0.8229 | 0.8039 | 0.5302 | 0.4001 |
| 5% | $1e^{-3}$ | 0.3904 | 0.3970 | 0.2202 | 0.2207 | - | - | - | - |
| | $1e^{-4}$ | 0.4105 | 0.4390 | 0.1968 | 0.1850 | 0.7351 | 0.5389 | 0.4789 | 0.2941 |
| | $1e^{-5}$ | 0.4404 | 0.4345 | 0.2069 | **0.1652** | 0.7377 | **0.0953** | 0.4258 | **0.0880** |
| | $1e^{-6}$ | 0.4212 | **0.3359** | 0.2030 | 0.2084 | 0.7238 | 0.4063 | 0.4364 | 0.3458 |
| | $1e^{-7}$ | 0.4433 | 0.4999 | 0.2115 | 0.2374 | 0.8277 | 0.7735 | 0.4990 | 0.4126 |
| 10% | $1e^{-3}$ | 0.4187 | 0.5360 | 0.2101 | 0.1843 | 0.7874 | 0.8453 | 0.4787 | 0.3305 |
| | $1e^{-4}$ | 0.4124 | 0.5314 | 0.1876 | 0.1338 | 0.7764 | 0.9376 | 0.4918 | 0.8172 |
| | $1e^{-5}$ | 0.3864 | 0.4585 | 0.2001 | **0.1215** | 0.7649 | **0.2791** | 0.4449 | **0.1057** |
| | $1e^{-6}$ | 0.4245 | **0.4211** | 0.2136 | 0.1623 | 0.7641 | 0.5214 | 0.4936 | 0.2777 |
| | $1e^{-7}$ | 0.4454 | 0.4872 | 0.2115 | 0.1843 | 0.8258 | 0.8307 | 0.5302 | 0.3139 |

Table 12: **UWC Tuning for the Batch Size of KL.** ↓ / ↑ indicate smaller / larger values are preferable.

| UWC | | Phi-1.5 | | | | Llama-2-7B | | | |
| | | ES-exact | | ES-perturb | | ES-exact | | ES-perturb | |
| setup | batch size | retain ↑ | unlearn ↓ | retain ↑ | unlearn ↓ | retain ↑ | unlearn ↓ | retain ↑ | unlearn ↓ |
|---|---|---|---|---|---|---|---|---|---|
| 1% | 4 | 0.4115 | 0.2904 | 0.1979 | **0.0000** | 0.7042 | 0.1082 | 0.4490 | 0.0154 |
| | 8 | 0.4232 | **0.1931** | 0.2005 | 0.1078 | 0.7241 | **0.0428** | 0.4791 | **0.0000** |
| | 12 | 0.4232 | 0.3238 | 0.2117 | 0.1126 | 0.7297 | 0.1952 | 0.4863 | 0.1043 |
| | 16 | 0.4232 | 0.2645 | 0.2136 | 0.1677 | 0.7249 | 0.1704 | 0.3928 | 0.0603 |
| | 20 | 0.4244 | 0.3531 | 0.1927 | 0.1412 | 0.7606 | 0.3072 | 0.3977 | 0.2072 |
| 5% | 4 | 0.4445 | 0.4022 | 0.2041 | 0.1272 | 0.7463 | 0.5809 | 0.4419 | 0.3627 |
| | 8 | 0.4404 | 0.4345 | 0.2069 | 0.1652 | 0.7377 | 0.0953 | 0.4258 | 0.0880 |
| | 12 | 0.3879 | 0.3352 | 0.2049 | **0.1432** | 0.6825 | **0.0590** | 0.4450 | **0.0604** |
| | 16 | 0.4211 | **0.2169** | 0.1882 | 0.1879 | 0.7836 | 0.5181 | 0.4496 | 0.1138 |
| | 20 | 0.4284 | 0.2514 | 0.1987 | 0.1879 | 0.7413 | 0.3749 | 0.4486 | 0.1443 |
| 10% | 4 | 0.3924 | 0.4736 | 0.2209 | 0.0826 | 0.7765 | 0.6994 | 0.5008 | 0.2605 |
| | 8 | 0.3864 | 0.4585 | 0.2001 | **0.1215** | 0.7649 | 0.2791 | 0.4449 | 0.1057 |
| | 12 | 0.4302 | **0.3358** | 0.2334 | 0.1621 | 0.7228 | **0.2287** | 0.4285 | **0.1071** |
| | 16 | 0.4424 | 0.4710 | 0.2225 | 0.1360 | 0.7557 | 0.3363 | 0.4769 | 0.1389 |
| | 20 | 0.3924 | 0.4340 | 0.2003 | 0.1238 | 0.7720 | 0.3990 | 0.4305 | 0.0927 |

Table 13: **UWC Tuning for the Unlearning Epochs of KL.** ↓ / ↑ indicate smaller / larger values are preferable.

| UWC | | Phi-1.5 | | | | Llama-2-7B | | | |
| | | ES-exact | | ES-perturb | | ES-exact | | ES-perturb | |
| setup | epochs | retain ↑ | unlearn ↓ | retain ↑ | unlearn ↓ | retain ↑ | unlearn ↓ | retain ↑ | unlearn ↓ |
|---|---|---|---|---|---|---|---|---|---|
| 1% | 1 | 0.4439 | 0.3368 | 0.2136 | 0.1551 | 0.8277 | 0.6284 | 0.4990 | 0.3444 |
| | 2 | 0.4223 | 0.2614 | 0.1942 | 0.1274 | 0.7370 | 0.2182 | 0.4560 | 0.2324 |
| | 3 | 0.4232 | **0.2033** | 0.2136 | **0.0571** | 0.8277 | **0.1029** | 0.4419 | 0.0403 |
| | 4 | 0.4232 | 0.2242 | 0.2005 | 0.1178 | 0.8277 | 0.1048 | 0.4435 | **0.0029** |
| 5% | 1 | 0.4393 | 0.2954 | 0.2192 | 0.2172 | 0.7418 | 0.5809 | 0.4563 | 0.3799 |
| | 2 | 0.4536 | **0.2224** | 0.2137 | **0.1386** | 0.7928 | 0.0231 | 0.4493 | 0.0144 |
| | 3 | 0.4268 | 0.2829 | 0.2276 | 0.1652 | 0.7496 | **0.0053** | 0.4420 | **0.0053** |
| | 4 | 0.4404 | 0.4395 | 0.2308 | 0.1652 | 0.7401 | 0.0053 | 0.4390 | 0.0620 |
| 10% | 1 | 0.4433 | **0.3974** | 0.2024 | 0.1360 | 0.7803 | **0.2163** | 0.4482 | 0.1076 |
| | 2 | 0.4424 | 0.4799 | 0.2004 | 0.1302 | 0.7939 | 0.3214 | 0.4828 | 0.1623 |
| | 3 | 0.4404 | 0.4575 | 0.2141 | **0.0715** | 0.7231 | 0.2479 | 0.4297 | **0.1071** |
| | 4 | 0.3944 | 0.4819 | 0.1813 | 0.1025 | 0.6989 | 0.2791 | 0.4487 | 0.1171 |

Table 14: **UWC Tuning for the Loss Selection of KL.** ↓ / ↑ indicate smaller / larger values are preferable.

| UWC | | Phi-1.5 | | | | Llama-2-7B | | | |
| | | ES-exact | | ES-perturb | | ES-exact | | ES-perturb | |
| setup | $q$ | retain ↑ | unlearn ↓ | retain ↑ | unlearn ↓ | retain ↑ | unlearn ↓ | retain ↑ | unlearn ↓ |
|---|---|---|---|---|---|---|---|---|---|
| 1% | 0.3 | 0.4934 | 0.4505 | 0.2513 | 0.1804 | 0.7958 | 0.7634 | 0.4832 | 0.4278 |
| | 0.5 | 0.4992 | 0.4506 | 0.2460 | 0.1709 | 0.7958 | 0.7634 | 0.4750 | 0.4217 |
| | 0.7 | 0.4620 | **0.3540** | 0.2443 | **0.1582** | 0.7900 | **0.6105** | 0.4656 | **0.3738** |
| 5% | 0.3 | 0.5786 | 0.5523 | 0.2544 | 0.1526 | 0.7509 | 0.6872 | 0.4694 | 0.3867 |
| | 0.5 | 0.5716 | 0.4859 | 0.2646 | 0.1625 | 0.6961 | 0.7309 | 0.4419 | 0.3757 |
| | 0.7 | 0.5766 | **0.2480** | 0.2492 | **0.1293** | 0.7080 | **0.3539** | 0.4299 | **0.2182** |
| 10% | 0.3 | 0.5879 | 0.5593 | 0.2466 | 0.2017 | 0.6860 | 0.6781 | 0.4463 | 0.3482 |
| | 0.5 | 0.5888 | 0.5262 | 0.2450 | 0.1951 | 0.6906 | 0.6914 | 0.4358 | 0.3621 |
| | 0.7 | 0.5909 | **0.4347** | 0.2462 | **0.1197** | 0.6984 | **0.4711** | 0.4249 | **0.1712** |

Table 15: **UWC Tuning for the Temperature Scaling of KL.** ↓ / ↑ indicate smaller / larger values are preferable.

| UWC | | Phi-1.5 | | | | Llama-2-7B | | | |
| | | ES-exact | | ES-perturb | | ES-exact | | ES-perturb | |
| setup | $\chi$ | retain ↑ | unlearn ↓ | retain ↑ | unlearn ↓ | retain ↑ | unlearn ↓ | retain ↑ | unlearn ↓ |
|---|---|---|---|---|---|---|---|---|---|
| 1% | 0.7 | 0.4590 | 0.1781 | 0.2532 | 0.1482 | 0.7175 | 0.4007 | 0.4238 | 0.2938 |
| | 0.9 | 0.4668 | 0.2389 | 0.2473 | 0.0955 | 0.7166 | 0.2006 | 0.4243 | 0.1892 |
| | 2.0 | 0.4853 | **0.0586** | 0.2517 | **0.0175** | 0.7327 | **0.0522** | 0.4304 | **0.0368** |
| 5% | 0.7 | 0.5824 | **0.3836** | 0.2447 | 0.1297 | 0.7057 | 0.3571 | 0.4154 | 0.2829 |
| | 0.9 | 0.6086 | 0.4067 | 0.2456 | 0.1189 | 0.7072 | 0.2896 | 0.4344 | 0.2556 |
| | 2.0 | 0.5776 | 0.5184 | 0.2473 | **0.0461** | 0.7018 | **0.1406** | 0.4362 | **0.0399** |
| 10% | 0.7 | 0.5927 | 0.5219 | 0.2495 | 0.1577 | 0.6847 | 0.6337 | 0.4314 | 0.3220 |
| | 0.9 | 0.5888 | **0.4786** | 0.2459 | 0.1546 | 0.6940 | 0.5619 | 0.4455 | 0.2464 |
| | 2.0 | 0.5881 | 0.4952 | 0.2493 | **0.1377** | 0.6851 | **0.0730** | 0.4278 | **0.0000** |

Table 16: Comparison between unlearning methods on 5% TOFU fictitious unlearning with UWC calibration across varying $\tau$. $\downarrow$ / $\uparrow$ indicate smaller / larger values are preferable. We primarily focus on the ES scores for unlearning (shaded), given that the ES scores for retention are calibrated.

| LLM | | Phi-1.5 | | | | Llama-2-7B | | | |
|---|---|---|---|---|---|---|---|---|---|
| | | ES-exact | | ES-perturb | | ES-exact | | ES-perturb | |
| setup | method | retain ↑ | unlearn ↓ | retain ↑ | unlearn ↓ | retain ↑ | unlearn ↓ | retain ↑ | unlearn ↓ |
| before unlearning | | 0.4433 | 0.5619 | 0.2115 | 0.2374 | 0.8277 | 0.7735 | 0.5302 | 0.4126 |
| $\tau = 0.4$ | GA | 0.2412 | **0.0286** | 0.1050 | 0.0623 | 0.3456 | 0.1217 | 0.1883 | 0.0333 |
| | GD | 0.2423 | 0.0400 | 0.1035 | **0.0000** | 0.3283 | **0.0000** | 0.2775 | **0.0000** |
| | KL | 0.2480 | 0.0533 | 0.1046 | **0.0000** | 0.3175 | **0.0000** | 0.1840 | **0.0000** |
| | PO | 0.2465 | 0.1927 | 0.0919 | 0.0794 | 0.3050 | 0.2116 | 0.1900 | 0.1617 |
| | NPO | 0.3020 | 0.1701 | 0.1566 | 0.0731 | 0.5378 | 0.1894 | 0.2736 | 0.1452 |
| | RMU | 0.2521 | 0.2505 | 0.1016 | 0.0576 | 0.3213 | 0.2065 | 0.1957 | 0.1578 |
| $\tau = 0.6$ | GA | 0.3213 | **0.0774** | 0.1443 | 0.0819 | 0.4725 | 0.1583 | 0.2693 | 0.1399 |
| | GD | 0.3517 | 0.0992 | 0.1434 | **0.0053** | 0.4768 | 0.0030 | 0.2850 | 0.0010 |
| | KL | 0.3620 | 0.1030 | 0.1538 | 0.0123 | 0.4700 | **0.0000** | 0.2747 | **0.0000** |
| | PO | 0.3978 | 0.4078 | 0.1555 | 0.1065 | 0.4848 | 0.3684 | 0.2875 | 0.2523 |
| | NPO | 0.4858 | 0.1992 | 0.1852 | 0.0824 | 0.6437 | 0.2811 | 0.3520 | 0.2050 |
| | RMU | 0.3871 | 0.3465 | 0.1646 | 0.0865 | 0.4900 | 0.3316 | 0.2700 | 0.1937 |
| $\tau = 0.9$ | GA | 0.5232 | **0.2021** | 0.2242 | 0.0825 | 0.7645 | 0.7010 | 0.4104 | 0.4800 |
| | GD | 0.5666 | 0.2200 | 0.2326 | **0.0753** | 0.7505 | 0.3300 | 0.4765 | 0.3200 |
| | KL | 0.5547 | 0.2153 | 0.2265 | 0.1080 | 0.7201 | **0.0944** | 0.4711 | **0.1580** |
| | PO | 0.5576 | 0.5358 | 0.2420 | 0.1634 | 0.7744 | 0.5493 | 0.4852 | 0.3596 |
| | NPO | 0.5691 | 0.2537 | 0.2269 | 0.1032 | 0.7210 | 0.1160 | 0.4753 | 0.2744 |
| | RMU | 0.5474 | 0.6038 | 0.2293 | 0.1352 | 0.7068 | 0.4004 | 0.4866 | 0.3741 |

