# OpenReview forum: "Towards Effective Evaluations and Comparisons for LLM Unlearning Methods"
_ICLR.cc/2025/Conference — ICLR 2025 Poster_

### Official Review · Reviewer_JEXC · 2024-10-31

**Soundness:** 3
**Presentation:** 2
**Contribution:** 2
**Rating:** 6
**Confidence:** 3

**Summary:**

The paper aims to answer the following research questions in machine unlearning: (1) What metric should be used in machine unlearning (2) what method is better in trading off unlearning and retention. The paper first presents a comparative study to understand what metric is robust (i.e., present a linear relationship before and after jailbreaking attacks), then proposes to use model mixing to for better calibration between retention and unlearning.

**Strengths:**

(1) The problem is well-motivated.

**Weaknesses:**

(1) For the metrics used for machine unlearning, The author suggests using Extraction Strength as the metrics since it exhibits linear relationship before and after different attacks. However, the evaluation is still weak to me. First of all, the attacks used in the paper are not the strongest jailbreaking attacks (like GCG attacks) and the authors also do not consider using an ensemble of strong attacks to elicit model’s knowledge, making the evaluation of the paper weak. Also, showing linear relationship before and after attack might not be a good way to measure machine unlearning before and after attack, especially if the model is backdoored or poisoned (e.g., a poisoned and backdoor model can hide their knowledge in their model parameters except when a trigger is presented or obfuscate the knowledge so that the metrics cannot used to detect unlearning). In short, there are no rigorous guarantees that using extraction strength is a valid metric under an adversarial scenario.

(2) The contribution is incremental. To me, the most novel part of the paper is that the authors use model mixing (aka model merging/souping) to control the trade-offs between retention and unlearning. All the other parts seem incremental and do not convey interesting empirical findings.

(3) The writing of the paper needs to be improved. A lot of useful details help to digest the results are shown in the appendix instead of the main text (e.g., attack methods) and the main text does not give enough concise summary to understand them. For example, I still cannot understand how we use Token Noising as an attack method (even the appendix does not give a good description of it). Also, the organization of the paper could be improved (e.g., lines 264-265 ask the reader to first read Section 6 for the experiment).

**Questions:**

Please provide response to the weaknesses.

---

> ### Author Response · Authors · 2024-11-19
> **Responses to Reviewer JEXC (part 1/2)**
>
> Many thanks for your constructive comments and suggestions! Please see our responses below.
>
> > **Q1**. For the metrics used for machine unlearning, The author suggests using Extraction Strength as the metrics since it exhibits linear relationship before and after different attacks. However, the evaluation is still weak to me. First of all, the attacks used in the paper are not the strongest jailbreaking attacks (like GCG attacks) and the authors also do not consider using an ensemble of strong attacks to elicit model’s knowledge, making the evaluation of the paper weak. Also, showing linear relationship before and after attack might not be a good way to measure machine unlearning before and after attack, especially if the model is backdoored or poisoned (e.g., a poisoned and backdoor model can hide their knowledge in their model parameters except when a trigger is presented or obfuscate the knowledge so that the metrics cannot used to detect unlearning). In short, there are no rigorous guarantees that using extraction strength is a valid metric under an adversarial scenario.
>
> **A1**. Many thanks for your comments! First, we would like to **clarify why linear relationships (or PCC) is appropriate for assessing the metric robustness**. As detailed in Appendix A, a proper metric that reflects the extent of knowledge parameterization should not be affected after attacking. In other words, the scores values before and after attacks should remain the same. However, it is also worth noting that, by the causal graph in Fig 4, **this conclusion holds only if the attacking methods change only superficial model behaviors while preserving the inherent knowledge**. For our considered 4 distinct types of red teaming scenarios, jail-breaking and probing strictly satisfy this criterion. Therefore, when considering only jail-breaking and probing, **we can ensure PCC approaches 1 and the slope of the linear fit is close to 1**, such that the scores before and after attacks lie in the identity line. As observed, in these criteria, ES is overall better than other metrics, especially for embed probing.
>
> However, relearning and input noising are also of practical interests, but they involve fine-tuning and in-context changes that may modify the inherent knowledge within the models. **Such modifications are likely to shift the distribution of scores, especially for relearning**. Therefore, to broaden the feasible range of applying attacking methods, we further loosen the above condition to test only the PPC scores. We apologize again for any confusion and will include the related discussion in our revision.
>
> **Moreover, we follow your suggestions to incorporate the GCG attacks**, the results are summarized as follows:
>
> PPL: 0.83 / 0.50; ES: 0.94 / 0.91; EM: 0.92 / 0.91.
>
> We also report the results that combine relearning and probing, which are two strongest attacking methods, the results are summarized as follows:
>
> PPL: 0.54 / 0.03; ES: 0.73 / 0.61; EM: 0.55 / 0.29.
>
> The results still show that ES is the most robust metric. We are always open to exploring additional attacking strategies. Many thanks for your wonderful suggestions!
>
> > **Q2**. The contribution is incremental. To me, the most novel part of the paper is that the authors use model mixing (aka model merging/souping) to control the trade-offs between retention and unlearning. All the other parts seem incremental and do not convey interesting empirical findings.
>
> **A2**. We sincerely appreciate your comments, but we respectfully disagree with your assessment regarding the lack of contributions of our work. We have succinctly outlined the key contributions of our work in the abstract, further distilled into two primary aspects—reliable metrics and reliable comparisons—in the **General Response**. Specifically, we challenge the reliability of commonly accepted metrics in the literature, such as those based on PPL and ROUGE scores, arguing that they are less effective than previously believed. Instead, we advocate for the ES as a more accurate measure of knowledge parameterization, supported by formal justifications. Additionally, we propose a calibration framework designed to simplify the challenges associated with comparing unlearning efficacy when dealing with dual objectives. This framework has spurred a series of experimental findings detailed in Section 6, which introduce new insights not previously discussed.
>
> To the best of our knowledge, **the aforementioned factors have not been discussed in prior research**. These insights contribute significantly to the community by enabling a more effective assessment and comparison of various unlearning methods. Kindly please correct us if there are any inaccuracies in our statements and we are really happy to further discuss with you about the contributions of our works. We will further highlight our contributions in our revision.

---

> ### Author Response · Authors · 2024-11-19
> **Responses to Reviewer JEXC (part 2/2)**
>
> > **Q3**. The writing of the paper needs to be improved. A lot of useful details help to digest the results are shown in the appendix instead of the main text (e.g., attack methods) and the main text does not give enough concise summary to understand them. For example, I still cannot understand how we use Token Noising as an attack method (even the appendix does not give a good description of it). Also, the organization of the paper could be improved (e.g., lines 264-265 ask the reader to first read Section 6 for the experiment).
>
> **A3**. We sincerely appreciate your suggestion! Regrettably, due to space constraints and our primary focus on discussing both appropriate evaluation metrics and calibration strategies, we have to place some of the important details to the appendix. We commit to dedicating more time to refining the organization and further emphasizing our key contributions in the main text. Additionally, we will allocate more space for discussions on experimental setups and results in the revised document.
>
> **For the reason why token noising can be viewed as an attacking method**, it is often the cases where overfitting occurs during unlearning. Thereby, token noise can lead to certain distribution shift, potentially causing the unlearning processes fail. Such vulnerabilities have been shown to lead to some successful attacks, such as that for EM. We will add the related discussions in our revision.

---

> ### Author Response · Authors · 2024-11-20
> **Looking forward to your reply**
>
> Dear Reviewer JEXC,
>
> Thank you for your great efforts in reviewing our work and for your insightful questions. We hope that our answers and general responses have helped to clarify the points discussed. Please let us know if there is anything else you require further information on or if there are any additional concerns you might have.
>
> Best regards,
>
> Authors of # 2049

---

> ### Author Response · Authors · 2024-11-24
> **Looking forward to your reply**
>
> Dear Reviewer JEXC,
>
> Many thanks for your great efforts in reviewing our works. Following your suggestions, we have conducted more experiments and further elaborated on our main contributions. We really hope that our answers can help to clarify.
>
> Since the discussion due is approaching, please let us know if there is any further comments or concerns you might have. We are also open to further discussions with you.
>
> Thank you very much!
>
> Best regards,
>
> Authors of # 2049

---

> ### Author Response · Authors · 2024-11-25
> **Looking forward to your reply**
>
> Dear Reviewer JEXC,
>
> We sincerely thank you for your efforts in reviewing our work! We believe we have addressed your initial concerns. As the deadline for discussion approaches, please let us know if you require any further clarifications. We are eager to continue the discussions with you.
>
> Looking forward to your response!
>
> Best regards,
>
> Authors of # 2049

---

> ### Comment · Reviewer_JEXC · 2024-11-25
> **Response after rebuttal**
>
> Thank you for your response. I still have some concerns over the attack included to to elicit model’s knowledge. A [concurrent work](https://arxiv.org/abs/2409.18025) (I know the work is too new so that it is unlikely you can compare with) has demonstrated fine-tuning attacks, orthogonalization, and some adaptive attacks like Enhanced GCG tailored for unlearning could easily recover the knowledge stored in the model. Thus, I tend to keep my score.

---

> > ### Author Response · Authors · 2024-11-26
> > **Further clarifications**
> >
> > Dear Reviewer JEXC,
> >
> > We sincerely thank you for your feedback and are very grateful for your suggestion to explore more attacking methods.
> >
> > We also checked your suggested paper, finding that finetuning has shown to be one of the strongest attack methods, and its much stronger version has already been included in our manuscript named as “relearn”. Therefore, it supports that **we have incorporated many strong attacking methods in our experiments**.
> >
> > Moreover, we have adapted the logit lens discussed in the suggested paper with its enhanced version named “probing”, already present in our manuscript. Following your suggestions, we have also presented the results of GCG attacks and the combinations of various attacking strategies in our previous responses. We will definitely integrate your suggestions to include enhanced GCG and orthogonalization in our revised manuscript, but we believe that **them cannot be used to judge that our current experimental analysis in Section 3 is insufficient**.
> >
> > We would also like to highlight that **our goal is not to test the unlearning efficacy of various methods, but to assess the reliability of different metrics**. Therefore, timely pursuing the strongest attacking methods is not our primary focus. Instead, we aim to apply well-recognized attacking methods to evaluate a broad spectrum of representative metrics to determine the most reliable ones.
> >
> > Based on our current experimental results in Section 3 as well as our further reported results in our original responses, **we have already had some unique and very informative findings**: We have discovered that widely used metrics like PPL and ROUGE may not be as reliable as previously thought. Conversely, ES, a less explored metric commonly used in membership inference, shows greater reliability in assessing unlearning efficacy.
> >
> > We believe that you had a slight misunderstanding of our work and, sadly, overlooked some of our unique contributions. We hope that our further clarifications will be taken into consideration. Thank you very much for your attention to these matters!
> >
> > Best regards,
> >
> > Authors of #2049

---

> > > ### Comment · Reviewer_JEXC · 2024-11-26
> > > **Response to Further Clarifications**
> > >
> > > Thank you for your clarification on the attack. I will raise my score.

---

> > > > ### Author Response · Authors · 2024-11-26
> > > > **Thank you**
> > > >
> > > > Thank you for your prompt feedback and for raising scores! We will definitely incorporate your suggestions to further refine our paper and include additional experiments in our revision. We greatly appreciate your support—it truly means a lot to us!

---

### Official Review · Reviewer_fSyL · 2024-11-04

**Soundness:** 2
**Presentation:** 1
**Contribution:** 2
**Rating:** 5
**Confidence:** 3

**Summary:**

The paper surveys the current popular evaluation methods towards LLM unlearning. It discuss the need to cover before retain and forget performances and its trade-offs, as well as its robustness against different attacks.

The paper concludes that Extraction Strength (ES) is the recommended method for evaluation. In addition, the paper also suggest Unlearning with Control (UWC) framework to reach the best trade-off between unlearning strength and retain performance.

**Strengths:**

The paper have a comprehensive view of different unlearning evaluation methods and approaches them in a systematic manner from robustness and utility trade-offs.

The paper proposes a novel approach unlearning with control to better calibrate the trade-off between unlearning effectiveness and retain performance with model-mixing, which is a simple but effective mechanism.

**Weaknesses:**

There is a lack of justification for selecting the metric: Why does the PCC measure the metrics' robustness again attacks? In Figure 2, the plot is characterized by the test static before and after the attack for different methods, models, and forget set ratio. Why should we assume there is a linear correlation among them? In addition, the paper uses TOFU as unlearning dataset/task, but does not survey the metric used in the TOFU paper (truth ratio).

Weak/unclear attack methods: it is unclear the what dataset the relearn attack in figure 2 is based on (from other section I would think it is TOFU, but it is not presented until late in the paper). The setup for relearning it not clear.

Although UWC seems like a method to manage the trade-off between unlearn and retain performance, it is not clear how it fits into unlearning evaluation pipeline.

**Questions:**

The statistic in figure 2 is confusing. How does model have lower score (which means better unlearning from the paper) after attacking than before?

---

> ### Author Response · Authors · 2024-11-19
> **Responses to Reviewer fSyL (part 1/2)**
>
> Many thanks for your constructive comments and suggestions! Please see our responses below.
>
> > **Q1**. There is a lack of justification for selecting the metric: Why does the PCC measure the metrics' robustness again attacks? In Figure 2, the plot is characterized by the test static before and after the attack for different methods, models, and forget set ratio. Why should we assume there is a linear correlation among them? In addition, the paper uses TOFU as unlearning dataset/task, but does not survey the metric used in the TOFU paper (truth ratio).
>
> **A1**. We apologize for any confusion that may arise. Here, we would like to further explain why PPC is a proper criterion for assessing the metric robustness.
>
> In Appendix A, we argue that an optimal metric should accurately reflect the extent of knowledge parameterization and not be affected by attacks. In other words, scores values before and after attacks should remain consistent. However, we want to further emphasize that, by the causal graph in Fig 4, **this criterion holds only if attack methods affect only superficial model behaviors while preserving the inherent knowledge intact**. For our considered 4 distinct types of red teaming scenarios, jail-breaking and probing critically satisfy this condition.
>
> Hence, when only focusing on jail-breaking and embed probing, we need to ensure **1)** PCC approaches 1 and **2)** the slope of the linear fit should be close to 1, thereby the scores before and after attacks align closely with the identity line. As observed, ES is overall better than other metrics, especially for embed probing. However, relearning and input noising also hold practical significance, but **they will alter the inherent knowledge within models via fine-tuning and in-context changes**. These two attacking processes can naturally lead to the shifts in the distributions of scores (especially for relearning). Therefore, we relax the conditions and focus solely on evaluating the PPC scores to include all four attacking scenarios of our interest.  We will refine the related discussion in our revision.
>
> **Now, we explain the reason why we did not explicitly include the truth ratio in our analysis**. Overall, we want to extract a series of **fundamental** metrics from existing scores, rather than continually adopting to the numerous new metrics that are proposed daily, which can be tedious and redundant (e.g., many proposed scores are based on ROUGE and PPL). The truth ratio can be derived from PPL, thereby having been covered in our analysis. However, we also agree with your opinion that the truth ratio has its unique characteristics, and we further conduct a series of experiments to assess its robustness. The PPC values obtained are as following:
>
> jail-breaking: 0.97 / 0.99; relearn: 0.56 / 0.09; probing: 0.83 / 0.02; noising: 0.99 / 0.99.
>
> As observed, **similar to PPL**, the truth ratio shows robustness against jail-breaking and noising. However, it does not outperform some of the more robust metrics when facing relearning attacks and embedding probing. We will incorporate the related discussion in our revision. Sincere thanks for your suggestions!
>
> > **Q2**. Weak/unclear attack methods: it is unclear the what dataset the relearn attack in figure 2 is based on (from other section I would think it is TOFU, but it is not presented until late in the paper). The setup for relearning it not clear.
>
> **A2**. Many thanks for your comment! Due to space limit, the detailed descriptions of the adopted attacking strategies have been allocated in the Appendix D. We adopt the original TOFU dataset for fine-tuning one epoch, using the negative log-likelihood as the objective. The Adam optimizer is adopted with the same learning rates as original fine-tuning. We will make the descriptions about the attacking setups clearer in our revision.

---

> ### Author Response · Authors · 2024-11-19
> **Responses to Reviewer fSyL (part 2/2)**
>
> > **Q3**. Although UWC seems like a method to manage the trade-off between unlearn and retain performance, it is not clear how it fits into unlearning evaluation pipeline.
>
> **A3**. The primary motivation behind UWC **is driven by the dual objectives of unlearning**: We should ensure effective removal of targeted knowledge and reliable retention of other, non-targeted knowledge, simultaneously.  However, achieving both goals often involves the trade-off between removal and retention.  For example, as illustrated in Fig 1, GA outperforms NPO in terms of removal, whereas NPO maintains better retention than GA. In this case, we cannot tell which method is overall better. These challenges motivate us to explore the concept of calibration, which aims to align overall performance across different models (i.e., ensuring retention). By ensuring consistent retention levels across different models / methods, we can then clearly compare their overall unlearning efficacy by **focusing solely in comparing their behaviors regarding the removal of specific knowledge**.
>
> **The benefits of calibration are substantial and twofold.**
>
> 1. **Fair comparisons**: Calibration can largely mitigate the difficulties of comparing unlearning efficacy across different methods, serving as the foundation for our experiments detailed in Sec 6. Therein, we have discovered new insights that were not mentioned previously. For example, after proper hyper-parameter tuning, GA-based methods have shown to be more effective than NPO; adapting batch sizes and implementing likelihood capping can further improve the overall efficacy of unlearning.
>
> 2. **Practical applications**: Beyond evaluations and comparisons, calibration also has benefits from an application perspective. It can be employed to recover the performance of non-targeted data for over-unlearned models. This is crucial for practical LLM applications, where unlearning must be conducted without impairing the normal model functionality. Given the current limited progress in retention, calibration can be employed post-unlearning to ensure the unlearned LLMs still useful.
>
> We will add the related discussions in our revision.
>
> > **Q4**. The statistic in figure 2 is confusing. How does model have lower score (which means better unlearning from the paper) after attacking than before?
>
> **A4**. Many thanks for your suggestion! As mentioned in the caption of Fig 2, the x- and y-axes denote the metric scores before and after attacks, respectively. The blue and orange dots / lines represent scores for retention and removal, respectively. As observed in all the attacking cases, the scores after attacking will either arise or remain unchanged, except for EM noising (indicating that EM is not a proper metric). We believe that your confusion may stem from a misunderstanding of the colors of the lines as indicating scores before and after attacks. We will clarify these descriptions in our revised manuscript to prevent any further misunderstanding.

---

> > ### Comment · Reviewer_fSyL · 2024-11-27
> >
> > > As observed in all the attacking cases, the scores after attacking will either arise or remain unchanged, except for EM noising (indicating that EM is not a proper metric).
> >
> > I am confused about this. It seems very clear that there are points are lie under the y=x line. Even a lot of fitted PCCs are smaller than 1. If the intuition is to keep the same score from before and after the attack, why do we need a linear fit? Should we just measure how close it is to the identity line? For PCC, if some methods is prone to attack but scales perfectly in a linear manner the PCC and still be at 1.
> >
> > In general I am a bit confused by the intuition of the paper. In addition, the relearning setup seems not be indicative of the unlearning robustness if all of TOFU dataset is used.

---

> > > ### Author Response · Authors · 2024-11-27
> > > **Further clarifications**
> > >
> > > Dear Reviewer fSyL,
> > >
> > > We sincerely thank you for your feedback! It looks that there are three points of your interests: **1)** the rationale for using PCC over identity line fitting in assessing metric robustness, **2)** the reason in adopting relearning as an attacking method, and **3)** the motivations and contributions of this paper. We are so sorry for these confusions and glad to further clarify these points.
> > >
> > > **PPC versus Identity Line Fitting**.
> > >
> > > Overall, we highlight that **criteria for assessing metric robustness depend on properties of attacking methods themselves**. **Formally**, the causal graph in Figure 4 indicates that **a reliable metric can maintain its value after attacks if the attacking method itself does not alter the knowledge within the inference process**. However, this condition is not satisfied by some attacking methods discussed in the literature. To make our assessment more adaptable to a broader range of attacking methods, we need to relax the strong criterion of identity line fitting. Two candidate criteria are correlation and PCC, both of which generalize identity line fitting scenarios.  However, **the requirements of PCC are more proper than correlation**, as the latter can refer to any type of relationship, either linear or non-linear. High correlation with non-linearity is generally not desirable for a reliable metric, because it will include many non-monotonic relationships that signify complete confusion in metric rankings before and after attacks, as well as non-smooth relationships that indicate strong impacts of the attacks on the metric. Considering the importance of linearity, we select PCC as our final criterion.
> > >
> > > **Intuitively**,  it is also not feasible to expect stable metric scores after strong attacks, especially for those that could alter the intrinsic knowledge within the inference process. **Consider the impact of feature noise as an example**: For an original input, “What gender is author Basil Mahfouz Al-Kuwaiti?”, if key terms like “gender” or “Basil” are randomly corrupted to unrelated words, such as “the”, the model output is likely to undergo significant changes. Therefore, it would be unrealistic to expect the metrics to retain their original values. Therefore, to cover a broad range of attacking methods, it is crucial to develop criteria that extend beyond exact matching.
> > >
> > > **Relearning as Attacks**.
> > >
> > > Relearning has been explored in previous works, which posits that **an effective unlearning method will impede the model’s ability to recover any previous-unlearned knowledge**. It supports the feasibility of incorporating targeted data for unlearning. Moreover, since **basic metrics will also be employed for assessing performance on other, non-targeted data**,  it is in our interest to observe how these metrics behave regarding knowledge that remains within models and is even enhanced. Therefore, involving non-targeted data during the unlearning process is also beneficial. We greatly appreciate your concerns and will include related discussions in our revised manuscript.
> > >
> > > **Motivations and Contributions**.
> > >
> > > Although new metrics are proposed daily, there remains a critical gap in comprehensive analysis regarding their reliability. We all know that **an unreliable metric can mislead the community towards superficial advancements, therefore underscoring the importance of rigorous metric assessment**. This concern **motivates** our discussion in Section 3, where we suggest a systematic way to validate the reliability of various metrics. Our analysis yields several new findings that we believe are **contributive**. For example, we raise concerns about the reliability of common-used metrics such as PPL and ROUGE, and identify ES, a less discussed metric, as a more reliable choice.
> > >
> > > We also contribute to the community by suggesting methods to fairly compare the efficacy between different unlearning approaches, where **we need to consider both removal and retention**. The challenges of pursuing these typically conflicting goals **motivate** us to propose model calibration, which can recover retention capabilities across various methods, thereafter solely assessing their capabilities of removal is enough for comparison. This calibration-based framework is **meaningful** and leads to several new findings, as detailed in Section 6. These includes the superior performance of GA-based methods over NPO after proper tuning, as well as enhancements in unlearning efficacy through adjusting batch sizes and likelihood capping.
> > >
> > > We sincerely hope that our clarifications above can address your concerns, and we always look forward to further discussions with you! Additionally, we want to further express our gratitude for your feedback, as we firmly believe that open discussions will not only enhance the quality of our paper but also help sustain an open meanwhile rigorous research community.
> > >
> > > Best regards,
> > >
> > > Authors of #2049

---

> ### Author Response · Authors · 2024-11-20
> **Looking forward to your reply**
>
> Dear Reviewer fSyL,
>
> Thank you for your great efforts in reviewing our work and for your insightful questions. We hope that our answers and general responses have helped to clarify the points discussed. Please let us know if there is anything else you require further information on or if there are any additional concerns you might have.
>
> Best regards,
>
> Authors of # 2049

---

> ### Author Response · Authors · 2024-11-24
> **Looking forward to your reply**
>
> Dear Reviewer fSyL,
>
> Many thanks for your great efforts in reviewing our works. Following your suggestions, we have conducted more experiments to strengthen our findings and clarified many technical details to enhance clarity. We really hope that our answers can help to clarify.
>
> Since the discussion due is approaching, please let us know if there is any further comments or concerns you might have. We are also open to further discussions with you.
>
> Thank you very much!
>
> Best regards,
>
> Authors of # 2049

---

> ### Author Response · Authors · 2024-11-25
> **Looking forward to your reply**
>
> Dear Reviewer fSyL,
>
> We sincerely thank you for your efforts in reviewing our work! We believe we have addressed your initial concerns. As the deadline for discussion approaches, please let us know if you require any further clarifications. We are eager to continue the discussions with you.
>
> Looking forward to your response!
>
> Best regards,
>
> Authors of # 2049

---

> ### Author Response · Authors · 2024-11-26
> **Looking forward to your reply**
>
> Dear Reviewer fSyL,
>
> As the deadline for discussion approaches, we are eager to know if there are any further questions or clarifications you require. We look forward to your response.
>
> Best regards,
>
> Authors of #2049

---

> ### Author Response · Authors · 2024-12-01
> **Looking forward to your reply**
>
> Dear Reviewer fSyL,
>
> Thank you for your follow-up! We are eager to know if our clarifications have addressed your concerns and look forward to further discussions.
>
> Best regards,
>
> Authors of #2049

---

### Official Review · Reviewer_8smY · 2024-11-09

**Soundness:** 4
**Presentation:** 3
**Contribution:** 3
**Rating:** 8
**Confidence:** 5

**Summary:**

This paper studies how best to evaluate LLM unlearning methods that aim to not leak the correct answers on a forget set while maintaining good performance on the rest of the training data. The literature has proposed a variety of evaluation metrics, and this paper first tackles which metric is most robust to information leaking attacks; they conclude ES, which was proposed with privacy attacks in mind, performs best in keeping the ranking before and after various attacks consistent. With ES, they then evaluated a generalized hyperparameter sweep for popular methods proposed in the literature, and conclude that methods have not significantly improved over the baseline of gradient ascent approaches. Narrowing why this is the case, they observe alternative methods either unlearn too strongly or not strongly enough to calibrate the performance on the retain set.

**Strengths:**

1) Extensive empirical study
2) Proposed method for improving calibration via a general hypermater boosts performance of baseline methods to seemingly SOTA
3) Mostly well-written

**Weaknesses:**

1) I found that certain parts of the draft could have been clearer about the benefits of model mixing. The draft does not discuss alternative calibration of retain performance, which naively could have also been done with just a sweep of the unlearning method hyperparameters. So at first I thought this was an empirical limitation. But after thinking about it I realized this is actually okay as the performance of calibration with just a hyperparameter sweep of the unlearning method is subsumed by the hyperparameter sweep including model mixing to calibrate (the former is just taking the hyperparameter in the latter where $\alpha  = 1$). Results in the appendix also suggest the hyperparameters for current unlearning methods are not sensitive enough to calibrate retain performance. In the questions section I ask clarifying questions about this and suggest edits to the text that could make this contribution clearer.

2) The empirical results are also focused on specific values of $\tau$, i.e., retain performance should be $95-90\%$ of the performance before unlearning. While I agree “high performance” seems to be the reasonable use-case, a more complete comparison without assuming specific $\tau$ values would also identify what $\tau$ range (if ever) NPO and other methods start performing well again. I believe it is implicit from the further study in the appendix that a “low” $\tau$ range might make other methods perform better, but I believe a more quantitative statement could make the empirical study more complete. I ask about this in the questions section.

3) (minor) A more general weakness, not specific to just this paper, is whether these performative unlearning goals for LLMs should also be generalization questions; if the goal is to not predict accurately for a set of samples, and those samples come from larger distribution, should we not be evaluating a “test” performance on the larger distribution? This is generally called concept unlearning in the literature. The field may still be far away from evaluating/studying such questions, but I raise it to clarify the setup studied in this paper is not the only setting for unlearning in LLMs. The authors might consider having more discussion on different formulations of unlearning before picking the setting they study and survey.

**Questions:**

I look forward to discussing the below with the authors, and am willing to increase my score given clarification of the questions below.

1) In the paragraph “Is MM Proper for Calibration” what seems missing is discussion comparing this to a simpler/baseline approach, such as just tuning the original hyperparameters of the unlearning method to calibrate. From the Appendix I got the impression that the hyperparameters of many of the unlearning methods do not give enough control to calibrate, is this correct? If so, discussion (and/or a figure illustrating this) would help the argument that MM is a better way to calibrate/boost the performance of unlearning methods.

2) Related to the above, in the “Hyper-parameter configurations” paragraph in the experiments, am I correct in understanding you do the hyperparameter sweep by evaluating the performance of the hyperparameters after also applying model mixing to calibrate the performance (this seems consistent with the appendix tables)? If so, you might consider adding somewhere in the paper that this hyperparameter sweep necessarily improves over the calibrated performance of not doing model mixing; this is as not doing model mixing is captured by just picking the best hyperparameters amongst those with $\alpha = 1$. I believe this adds conceptual understanding to why model mixing is necessarily better.

3) On the discussion of why NPO and other methods do no work in the experiments section, do you have results for what $\tau$ values (if at all) other methods start performing well? I’m imagining a figure where the x-axis is calibrated $\tau$ level, and the y-axis is forget performance, and a line for each method such that when the line is the lowest explains for what $\tau$ each method is best. Alternatively, some more detailed/explicit discussion for what $\tau$ levels other methods might perform better could help give a more complete picture for when other methods work well. I agree the high $\tau$ range seems most practical, but I think the above can be done easily given results in the appendix and can add to the empirical study.

4) (minor) model mixing seems very related to the literature on model merging; you might consider surveying the literature more broadly as a reference for the method, and when/why it works.

5) (minor) If computationally feasible, could error bars/standard deviation be reported for some of the main tables; I am less concerned given the many settings tested, but this could add to the rigor of the empirical findings.

---

> ### Author Response · Authors · 2024-11-19
> **Responses to Reviewer 8smY (part 1/4)**
>
> Many thanks for your great support and constructive comments! Please see our responses below.
>
> > **Q1**. In the paragraph “Is MM Proper for Calibration” what seems missing is discussion comparing this to a simpler/baseline approach, such as just tuning the original hyperparameters of the unlearning method to calibrate.
>
> **A1**. Many thanks for your wonderful suggestions! We had not considered this aspect previously, but we totally agree with your view that hyper-parameter tuning can indeed be considered as a form of calibration. To highlight the superiority of MM, we would like to emphasize that a proper calibration method should ensure the control is applied **in a noticeable yet smooth manner**. However, as observed in Appendix H regarding hyper-parameter tuning, the model behaviors are quite sensitive to the choices of hyper-parameters, and we do not achieve the desired level of recovery in overall performance for many cases of hyper-parameter search. In contrast, in Fig 3, the control exerted by MM over model behaviors is notably smooth.
>
> Additionally, **conducting calibration through hyper-parameter tuning is too method-specific**, which is not general for constructing a reliable evaluation framework. The computational costs associated with tuning hyper-parameters are also prohibitively high. By contrast, MM can be applied post-unlearning across various methods without incurring the additional costs associated with re-unlearning. Therefore, we believe MM is more general, reliable, flexible, and efficient than hyper-parameter tuning in calibration. We will add the related discussions in our revision.
>
> > **Q2**. Related to the above, in the “Hyper-parameter configurations” paragraph in the experiments, am I correct in understanding you do the hyperparameter sweep by evaluating the performance of the hyperparameters after also applying model mixing to calibrate the performance (this seems consistent with the appendix tables)? If so, you might consider adding somewhere in the paper that this hyperparameter sweep necessarily improves over the calibrated performance of not doing model mixing; this is as not doing model mixing is captured by just picking the best hyperparameters amongst those with \alpha=1.
>
> **A2**. Yes, your understanding is accurate. Moreover, we would like to highlight that **calibration eases the comparison of overall efficacy across different methods, especially when some methods excel at data removal while others outperform in data retention (cf. Fig 1)**. In line with your suggestions and to show that these scenarios are common in practice, we report the forget05 results before calibration in the following as a case study.  As observed, the removal efficacy of GD exceeds that of NPO, while its retention performance is inferior. **Since both removal and retention are crucial to us, it is hard to conclude which method is superior overall**. We will add more results in our revision, many thanks for your suggestion!
>
> | Model  |     Phi-1.5     |      Phi-1.5     |      Phi-1.5      |       Phi-1.5      |    Llama-2-7B   |    Llama-2-7B    |     Llama-2-7B    |     Llama-2-7B     |
> |--------|:---------------:|:----------------:|:-----------------:|:------------------:|:---------------:|:----------------:|:-----------------:|:------------------:|
> | Metric | PS exact retain | PS exact unlearn | PS perturb retain | PS perturb unlearn | PS exact retain | PS exact unlearn | PS perturb retain | PS perturb unlearn |
> | GA     |      0.000     |      0.000      |       0.000      |       0.000       |     0.000      |      0.000      |      0.000       |       0.000       |
> | GD     |      0.129     |      0.000      |       0.099      |       0.000       |      0.326     |      0.000      |       0.220      |       0.000       |
> | KL     |      0.110     |      0.000     |       0.100      |       0.000       |      0.221     |      0.000      |       0.166      |       0.000       |
> | PO     |      0.059     |      0.070      |       0.066      |       0.064       |      0.246     |      0.116      |       0.190      |       0.153       |
> | NPO    |      0.136     |      0.087      |       0.099      |       0.072       |      0.499     |      0.089      |       0.305      |       0.078       |
> | RMU    |     0.000      |      0.000      |      0.000       |       0.000       |     0.000      |      0.000      |      0.000       |       0.000       |
>
> Beyond evaluations, **calibration also has benefits from the application perspective, which can be employed to recover the common performance for unlearned models**. This is crucial for practical LLM applications, where unlearning must occur without compromising the normal model functionality. Given the current limited progress in retention, model mixing (or calibration) also holds practical significance.  We will add the related discussions in our revision.

---

> ### Author Response · Authors · 2024-11-19
> **Responses to Reviewer 8smY (part 2/4)**
>
> > **Q3**. On the discussion of why NPO and other methods do no work in the experiments section, do you have results for what $\tau$ values (if at all) other methods start performing well? I’m imagining a figure where the x-axis is calibrated  $\tau$ level, and the y-axis is forget performance, and a line for each method such that when the line is the lowest explains for what $\tau$ each method is best. Alternatively, some more detailed/explicit discussion for what  $\tau$ levels other methods might perform better could help give a more complete picture for when other methods work well. I agree the high $\tau$ range seems most practical, but I think the above can be done easily given results in the appendix and can add to the empirical study.
>
> **A3**. Many thanks for your suggestions! Please see below for the calibrated results under 5% unlearning setups across different values of $\tau$.
> | $\tau$=0.4 |     Phi-1.5     |      Phi-1.5     |      Phi-1.5      |       Phi-1.5      |    Llama-2-7B   |    Llama-2-7B    |     Llama-2-7B    |     Llama-2-7B     |
> |:----------:|:---------------:|:----------------:|:-----------------:|:------------------:|:---------------:|:----------------:|:-----------------:|:------------------:|
> |   Metric   | PS exact retain | PS exact unlearn | PS perturb retain | PS perturb unlearn | PS exact retain | PS exact unlearn | PS perturb retain | PS perturb unlearn |
> |     GA     |      0.241     |      0.028      |       0.105      |       0.062       |      0.345     |      0.121      |       0.188      |       0.033       |
> |     GD     |      0.242     |      0.040      |       0.103      |       0.000       |      0.328     |      0.000      |       0.227      |       0.000       |
> |     KL     |      0.248     |      0.053      |       0.104      |       0.000       |      0.317     |      0.000      |       0.184      |       0.000       |
> |     PO     |      0.246     |      0.192      |       0.091      |       0.079       |      0.305     |      0.211      |       0.190      |       0.161       |
> |     NPO    |      0.302     |      0.170      |       0.156      |       0.073       |      0.537    |      0.189      |       0.273      |       0.145       |
> |     RMU    |      0.252     |      0.250      |       0.101      |       0.057       |      0.321     |      0.206      |       0.195      |       0.157       |
>
> | $\tau$=0.6 |     Phi-1.5     |      Phi-1.5     |      Phi-1.5      |       Phi-1.5      |    Llama-2-7B   |    Llama-2-7B    |     Llama-2-7B    |     Llama-2-7B     |
> |:----------:|:---------------:|:----------------:|:-----------------:|:------------------:|:---------------:|:----------------:|:-----------------:|:------------------:|
> |   Metric   | PS exact retain | PS exact unlearn | PS perturb retain | PS perturb unlearn | PS exact retain | PS exact unlearn | PS perturb retain | PS perturb unlearn |
> |     GA     |      0.321     |      0.077      |       0.144      |       0.081       |      0.472     |      0.158      |       0.269      |       0.139       |
> |     GD     |      0.351     |      0.099      |       0.143      |       0.005       |      0.476     |      0.003      |       0.285      |       0.001       |
> |     KL     |      0.362     |      0.103      |       0.153      |       0.012       |      0.470     |      0.000      |       0.274      |       0.000       |
> |     PO     |      0.397     |      0.407      |       0.155      |       0.106       |      0.484     |      0.368      |       0.287      |       0.252       |
> |     NPO    |      0.485     |      0.199      |       0.185      |       0.082       |      0.643    |      0.281      |       0.352      |       0.205       |
> |     RMU    |      0.387     |      0.346      |       0.164      |       0.086       |      0.490     |      0.331      |       0.270      |       0.193       |

---

> ### Author Response · Authors · 2024-11-19
> **Responses to Reviewer 8smY (part 3/4)**
>
> | $\tau$=0.9 |     Phi-1.5     |      Phi-1.5     |      Phi-1.5      |       Phi-1.5      |    Llama-2-7B   |    Llama-2-7B    |     Llama-2-7B    |     Llama-2-7B     |
> |:----------:|:---------------:|:----------------:|:-----------------:|:------------------:|:---------------:|:----------------:|:-----------------:|:------------------:|
> |   Metric   | PS exact retain | PS exact unlearn | PS perturb retain | PS perturb unlearn | PS exact retain | PS exact unlearn | PS perturb retain | PS perturb unlearn |
> |     GA     |      0.523     |      0.202      |       0.224      |       0.082       |      0.764     |      0.701      |       0.410      |       0.480       |
> |     GD     |      0.566     |      0.220      |       0.232      |       0.075       |      0.750     |      0.330      |       0.476      |       0.320       |
> |     KL     |      0.554     |      0.215      |       0.226      |       0.108       |      0.720     |      0.094      |       0.471      |       0.158       |
> |     PO     |      0.557     |      0.535      |       0.242      |       0.163       |      0.774     |      0.549      |       0.485      |       0.359       |
> |     NPO    |      0.569     |      0.253      |       0.226      |       0.103       |      0.721     |      0.116      |       0.475      |       0.274       |
> |     RMU    |      0.547     |      0.603      |       0.229      |       0.135       |      0.706     |      0.400      |       0.486      |       0.374       |
>
> As we can see, GA-based methods can still beat NPO and other methods. We thank for your wonderful suggestion and we will add the related discussion in our revision.
>
> > **Q4**. (minor) A more general weakness, not specific to just this paper, is whether these performative unlearning goals for LLMs should also be generalization questions; if the goal is to not predict accurately for a set of samples, and those samples come from larger distribution, should we not be evaluating a “test” performance on the larger distribution? This is generally called concept unlearning in the literature. The field may still be far away from evaluating/studying such questions, but I raise it to clarify the setup studied in this paper is not the only setting for unlearning in LLMs. The authors might consider having more discussion on different formulations of unlearning before picking the setting they study and survey.
>
>
> **A4**. We totally agree with your opinion. Your mentioned perspective is also supported by previous works that discuss about the **scope of unlearning**. Therein, they mentioned that, depending on the application scenarios, the unlearning goal may involve either removing a specific set of data or more general concepts/knowledge. While the TOFU benchmark offers a set of rephrased data for testing generalization (adopted in our evaluations as ES-perturb), it does not fully enable us to ablate or explore how to control the scope of unlearning. It is a fascinating yet challenging direction, of which the studies should involve collaboration on new unlearning benchmarks, metrics, and methods. We will certainly include a detailed discussion about it in our revised manuscript. Many thanks for your wonderful comments!
>
> [1] Sijia et al. Rethinking Machine Unlearning for Large Language Models.
>
> > **Q5**. (minor) model mixing seems very related to the literature on model merging; you might consider surveying the literature more broadly as a reference for the method, and when/why it works.
>
> **A5**. We fully concur with your perspective that our model mixing is closely related to model merging, a concept previously explored in the robust fine-tuning, model editing, and federated learning literature. Prior empirical studies have demonstrated that averaging over a set of models tends to preserve the properties of each [2,3]. Additionally, theoretical research [4] has suggested that the property of weight disentanglement, which posits that averaging weights is akin to an ensemble of models, is crucial for the success of model mixing (or model merging). We will add the related discussion in our revision.
>
> [2] Wortsman et al. Robust Fine-tuning of Zero-shot Models.
>
> [3] Ilharco et al. Editing Models with Task Arithmetic.
>
> [4] Ortiz-Jimenez et al. Task Arithmetic in the Tangent Space.

---

> ### Author Response · Authors · 2024-11-19
> **Responses to Reviewer 8smY (part 4/4)**
>
> > **Q6**. (minor) If computationally feasible, could error bars/standard deviation be reported for some of the main tables; I am less concerned given the many settings tested, but this could add to the rigor of the empirical findings.
>
> **A6**. Thank you for your suggestion! We report the mean and standard deviation (mean + std) results under the 5% forget setup as follows. All experiments were conducted five times using different random seeds. Our observations indicate that GA-based methods continue to outperform NPO, consistent with our original findings in Table 1. We will include additional results and analysis in our revision.
>
> | Model  |     Phi-1.5     |      Phi-1.5     |      Phi-1.5      |       Phi-1.5      |    Llama-2-7B   |    Llama-2-7B    |     Llama-2-7B    |     Llama-2-7B     |
> |--------|:---------------:|:----------------:|:-----------------:|:------------------:|:---------------:|:----------------:|:-----------------:|:------------------:|
> | Metric | PS exact retain | PS exact unlearn | PS perturb retain | PS perturb unlearn | PS exact retain | PS exact unlearn | PS perturb retain | PS perturb unlearn |
> | GA     |  0.438 + 0.020  |   0.298 + 0.015  |   0.224 + 0.023   |    0.245 + 0.019   |  0.778 + 0.018  |   0.713 + 0.015  |   0.394 + 0.016   |    0.476 + 0.008   |
> | GD     |  0.418 + 0.018  |  0.425 + 0.012   |   0.188 + 0.027   |    0.005 + 0.014   |  0.753 + 0.015  |   0.329 + 0.018  |   0.453 + 0.023   |   0.318 + 0.013    |
> | KL     |  0.404 + 0.015  |   0.386 + 0.014  |   0.188 + 0.021   |    0.165 + 0.013   |  0.740 + 0.017  |   0.102 + 0.012  |   0.494 + 0.009   |    0.158 + 0.014   |
> | PO     |  0.416 + 0.017  |   0.458 + 0.007  |   0.210 + 0.010   |    0.227 + 0.011   |  0.784 + 0.006  |   0.560 + 0.013  |   0.488 + 0.007   |    0.350 + 0.004    |
> | NPO    |  0.435 + 0.005  |   0.363 + 0.008  |   0.184 + 0.000   |    0.151 + 0.005   |  0.721 + 0.003  |   0.110 + 0.004  |   0.480 + 0.001   |    0.279 + 0.003   |
> | RMU    |  0.421 + 0.021  |   0.437 + 0.013  |   0.194 + 0.017   |    0.215 + 0.013   |  0.724 + 0.013  |   0.393 + 0.010  |   0.450 + 0.012   |    0.393 + 0.015   |

---

> ### Comment · Reviewer_8smY · 2024-11-19
> **Response to Authors**
>
> I thank the authors for their response! I believe the paper will have an impact on the field, and given the clarification to my understanding, I am raising both my score and confidence. In particular, I’m stating confidence 5 as to normalize with the other reviewers given my understanding of their reviews. I elaborate below.
>
> To elaborate on my stance, I have also read the other reviews, and disagree with their complaints about novelty and impact. A core issue in comparing unlearning methods is that unlearning methods do not equally weigh retain and forget performance (even if more compute is spent into calibration), and this method resolves this simply with seemingly great success. It seems the other reviewers do not believe the method is novel because of its simplicity, but such an argument seems ill-grounded given the method's effectiveness and lack of previous literature on the problem. Furthermore, as now pointed out (which I hope will be further elaborated on in the final draft), the obvious approach of just hyperparameter tuning for calibration does not work and is also more computationally expensive.
>
> Another core issue in the field is that there are many metrics for unlearning, and this paper gives empirical evidence towards working with a single metric. Complaints were raised on rigor, but the claim in the paper is to have a metric that best preserves ranking, which is what the correlation metric does (though spearman may be better than pearson, but given the plots linear seems appropriate). Similar methodology was already used in published empirical studies in security conferences (with the difference being a focus on unlearning for privacy legislation) [1]. Another point was raised that poisoning and backdooring attacks would not fit such tests, but it is already known such attacks are computationally undetectable [2] and hence out-of-scope for the threat model of all current unlearning methodologies (which is implicitly assuming computational feasibility of unlearning). Thinking of the broader field, tests will need to continue to be made as more attacks are known, but this paper presents a case study and methodology showing great promise on how to go about these comparisons.
>
> [1] “Anvith Thudi, Gabriel Deza, Varun Chandrasekaran, and Nicolas Papernot. Unrolling sgd: Understanding factors influencing machine unlearning. In 2022 IEEE 7th European Symposium on Security and Privacy (EuroS&P)”
>
> [2] “Goldwasser, Shafi, et al. "Planting undetectable backdoors in machine learning models." 2022 IEEE 63rd Annual Symposium on Foundations of Computer Science (FOCS). IEEE, 2022.”

---

> > ### Author Response · Authors · 2024-11-20
> > **Thank you**
> >
> > Dear Reviewer 8smY,
> >
> > We sincerely appreciate your great support and insightful comments, which mean a lot to us! Also, following your suggestions, we will highlight the reasons why hyper-parameter tuning is inappropriate for calibration and further emphasize our contributions towards identifying and evaluating reliable unlearning metrics in our revision.
> >
> > Thank you once again for your valuable time and insights!
> >
> > Best regards,
> >
> > Authors of # 2049

---

### Official Review · Reviewer_taYS · 2024-11-10

**Soundness:** 2
**Presentation:** 2
**Contribution:** 2
**Rating:** 3
**Confidence:** 4

**Summary:**

The paper tested 4 popular LLM unlearning methods on the TOFU benchmark dataset and observed that there are various tradeoffs (e.g. GA unlearns better while NPO retains better). Then it motivates the authors to propose a new metric which is based on the mixing the model weights between the unlearned and original model. The reason, as far as I understand, is to achieve smooth control on the extent of unlearning. The evaluation first finds the model mixing ratio that would achieve the evaluation score (e.g. ES) on the retain set, and then uses it to calibrate unlearning metric.

**Strengths:**

S1: LLM unlearning and evaluation are important problems

**Weaknesses:**

W1: Lack of technical contribution: I think most people working in this area would agree we need more metrics and benchmark datasets. However, this paper though goes into that direction, does not really provide enough meaningful and technical contribution in my view. The paper basically tried 4 popular unlearning methods on the TOFU datasets while proposing a calibration framework (See W2). This can mostly be done in leaderboard or in a measurement paper rather than a technical paper. And findings on metric tradeoffs are mostly not surprising.

W2: Lack of justification of the calibration framework: The calibration metric lacks justification. Figure 3 seems to only tell us ES is monotonically increasing with mixing factor $\alpha$. However, the key question in proposing a metric is to ask: What does this metric measure? Does it measure the right thing? In this case, I cannot see clearly the impact of mixing model weights on the overall unlearning effectiveness. The only justification I can find is (1) being inspired by the literature on parameter disentanglement, but why is it related to unlearning? (2) The vague observation in Figure 3. Can authors explain why mixing model weights can help calibrate the unlearning metric other than just the empirical observation between ES and $\alpha$ in Figure 3? After all, this can well be a spurious correlation. In other words, why would the community trust this calibration in evaluation?

W3: Lack of organizing clarify: The paper spends the first 6 pages motivating the calibration framework, and only 1 page introducing it and 1 page for experiments. I think people who in this area already know most of the points in the first 6 pages, it'd better to shorten it and get to the point more directly and spent more content on introducing the key idea of calibration and include more justification,

**Questions:**

See W2.

---

> ### Author Response · Authors · 2024-11-19
> **Responses to Reviewer taYS (part 1/3)**
>
> Many thanks for your constructive comments and suggestions! Please see our responses below.
>
> > **Q1**. Lack of technical contribution: I think most people working in this area would agree we need more metrics and benchmark datasets. However, this paper though goes into that direction, does not really provide enough meaningful and technical contribution in my view. The paper basically tried 4 popular unlearning methods on the TOFU datasets while proposing a calibration framework (See W2). This can mostly be done in leaderboard or in a measurement paper rather than a technical paper. And findings on metric tradeoffs are mostly not surprising.
>
> **A1**. We sincerely appreciate your comments, but we respectfully disagree with your assessment regarding the lack of contributions of our work. Your view that this manuscript is not a technical paper outlooks our systematical comparisons of various unlearning metrics and methods (cf. Sec 3-5) as well as our justification for the rationale behind them (cf. Appendix A, Sec 4, and Appendix F-G). Additionally, our experiments cover 6 (instead of 4) representative baselines as well as a set of practical tricks that can further improve the effectiveness of existing works.
>
> We have succinctly outlined the key contributions of our work in the abstract, further distilled into two primary aspects—reliable metrics and reliable comparisons—in the **General Response**. Specifically, we challenge the reliability of commonly accepted metrics in the literature, such as those based on PPL and ROUGE scores, arguing that they are less effective than previously believed. Instead, we advocate for the ES as a more accurate measure of knowledge parameterization, supported by formal justifications. Additionally, we propose a calibration framework designed to simplify the challenges associated with comparing unlearning efficacy when dealing with dual objectives. This framework has spurred a series of experimental findings detailed in Section 6, which introduce new insights not previously discussed.
>
> **These factors contribute significantly to the community by enabling a more effective assessment and comparison of various unlearning methods**. Kindly please correct us if there are any inaccuracies in our statements and we are really happy to further discuss with you about the contributions of our works. Additionally, we will follow your suggestions (especially from Q3) to refine our paper to better highlight our contributions.

---

> ### Author Response · Authors · 2024-11-19
> **Responses to Reviewer taYS (part 2/3)**
>
> > **Q2**. Lack of justification of the calibration framework: The calibration metric lacks justification. Figure 3 seems to only tell us ES is monotonically increasing with mixing factor. However, the key question in proposing a metric is to ask: What does this metric measure? Does it measure the right thing? In this case, I cannot see clearly the impact of mixing model weights on the overall unlearning effectiveness. The only justification I can find is (1) being inspired by the literature on parameter disentanglement, but why is it related to unlearning? (2) The vague observation in Figure 3. Can authors explain why mixing model weights can help calibrate the unlearning metric other than just the empirical observation between ES and  in Figure 3? After all, this can well be a spurious correlation. In other words, why would the community trust this calibration in evaluation?
>
> **A2**. We apologize for any confusion that may arise. To begin with, we would like to **highlight the motivation for the concept of calibration**. For a practical unlearning method, we should pursue both effective removal and reliable retention. However, **there is typically a trade-off between these two objectives**. It complicates comparisons among unlearning methods, as one method might excel in retention while another is better at removal. Consequently, it becomes challenging to determine which method is superior overall.  To address this, we propose a novel strategy: **calibration**. By aligning the retention performance across different methods, we can focus on comparison solely on their removal efficacy, **overcoming the notorious trade-off issues and ensuring a more fair comparison**.
>
> The question now is what ensures an effective calibration approach. Ideally, **it should manage model performance smoothly**, displaying monotonic and stable changes in terms of both retention and removal. By achieving this, we can align different models in a stable manner (aligning retention performance) meanwhile ensure a fair comparison of removal efficacy (as the drop in removal efficacy is also fairly controllable).
>
> **The primary purpose of Fig 3** is to demonstrate that model mixing can achieve the goal of smooth control. The horizonal and vertical axes represent the controlling parameter $\alpha$ within model mixing and the value of ES score (cf. the caption of Fig 3), respectively. ES scores are used to measure the performance with respect to both retention (blue) and removal (orange). We observe that across different values of $\alpha$, **the performance for both retention and removal changes in a noticeable yet smooth manner**. Thereby, we conclude that model mixing is appropriate for calibration purposes. It appears that your misunderstanding may arise because we placed our descriptions of the definitions for the axes in the caption. We will clarify Fig 3 further in our revision.
>
> We would like to further highlight that Fig 3 is our justification for the reliability of our evaluation framework. **There are no spurious factors within our calibration method, as it relies solely on the controlling parameter $\alpha$**. We are looking forwards to further discussions on the rationale and contributions of model mixing and calibration. We also welcome any additional concerns or question you may have!!

---

> > ### Author Response · Authors · 2024-12-01
> > **Looking forward to your reply**
> >
> > Dear Reviewer taYS,
> >
> > Thank you for your great efforts in reviewing our work and for your insightful questions. We believe we have addressed many of your concerns, please let us know if there is anything else we can further discuss. Thank you!
> >
> > Best regards,
> >
> > Authors of # 2049

---

> ### Author Response · Authors · 2024-11-19
> **Responses to Reviewer taYS (part 3/3)**
>
> > **Q3**. Lack of organizing clarify: The paper spends the first 6 pages motivating the calibration framework, and only 1 page introducing it and 1 page for experiments. I think people who in this area already know most of the points in the first 6 pages, it'd better to shorten it and get to the point more directly and spent more content on introducing the key idea of calibration and include more justification.
>
> **A3**. Many thanks for your suggestions. As aforementioned in A1, our contributions are twofold. In the first 6 pages, we allocate 2 pages to the abstract and introduction, 1 pages to the necessary background, 2 pages to our first contribution on finding a reliable unlearning metric, and 1 pages for our second contribution on proposing the calibration framework. Therefore, the first 6 pages encompass a lot of new discussions. We sincerely appreciate your feedback and agree that there is still a large room to polish the organization of our manuscript. Following your suggestions, we will condense the introduction and preliminary sections, and allocate more space to elaborate on the calibration process and the experimental results. Thanks again for your suggestions!

---

> ### Author Response · Authors · 2024-11-20
> **Looking forward to your reply**
>
> Dear Reviewer taYS,
>
> Thank you for your great efforts in reviewing our work and for your insightful questions. We really hope that our answer and the general responses can help to clarify. Please let us know if there is anything else you need further information on or any additional concerns you might have.
>
> Best regards,
>
> Authors of # 2049

---

> ### Author Response · Authors · 2024-11-24
> **Looking forward to your reply**
>
> Dear Reviewer taYS,
>
> Thank you for your great efforts in reviewing our work and for your insightful questions. Following your suggestions, we have elaborated on our main contributions and clarified technical details for enhanced clarity. We really hope that our answers can help to clarify.
>
> Since the discussion due is approaching, please let us know if there is anything else you need further information on or any additional concerns you might have.
>
> Best regards,
>
> Authors of # 2049

---

> ### Author Response · Authors · 2024-11-25
> **Looking forward to your reply**
>
> Dear Reviewer taYS,
>
> We sincerely thank you for your efforts in reviewing our work! We believe we have addressed your initial concerns. As the deadline for discussion approaches, please let us know if you require any further clarifications. We are eager to continue the discussions with you.
>
> Looking forward to your response!
>
> Best regards,
>
> Authors of # 2049

---

> ### Author Response · Authors · 2024-11-26
> **Looking forward to your reply**
>
> Dear Reviewer taYS,
>
> As the deadline for discussion approaches, we are eager to know if there are any further questions or clarifications you require. We look forward to your response.
>
> Best regards,
>
> Authors of #2049

---

> ### Author Response · Authors · 2024-11-27
> **Looking forward to your reply**
>
> Dear Reviewer taYS,
>
> Kindly please take a moment to review our feedback at your convenience. We will always await your responses and are looking forward to further discussions!
>
> Best regards,
>
> Authors of #2049

---

> ### Author Response · Authors · 2024-11-28
> **Looking forward to your reply**
>
> Dear Reviewer taYS,
>
> We are still awaiting your further feedback and would appreciate any additional comments or questions that you might have.
>
> Best regards,
>
> Authors of #2049

---

### Author Response · Authors · 2024-11-19
**General Response to All Reviewers**

We are grateful to all the reviewers for their time and insightful feedback. We appreciate the reviewers' recognition of the importance of the problems we have addressed in LLM unlearning (taYS, JEXC) as well as the novelty (8smY, fSyL) of our proposed assessment methods. The reviewers have also acknowledged our comprehensive experiments across a variety of metrics and unlearning methods (8smY, fSyL).

We also sincerely appreciate the concerns some reviewers have raised regarding the key contributions and the uniqueness of our manuscript. We apologize for any confusion and thank you for the opportunity to clarify these points. We have briefly summarized our uniqueness in the abstract, and we are glad to provide a more detailed summary, organized into two main points, as follows:

1.	**Reliable Metrics**: Although many metrics have been proposed for assessing LLM unlearning, there is still a lack of comprehensive analysis of these metrics’ reliability. We address this gap by stating that **a proper metric should characterize the knowledge parameterization within models**. We also suggest **a systematic method to validate  the appropriateness of a metric via red team attacks**. Its rationale has been rigorously validated in Appendix A, alongside with criteria for assessing the suitability of attacking methods. We identify a set of basic metrics widely adopted in previous works, and **challenge the reliability of commonly used metrics, such as PPL and ROUGE. Also, we find a less discussed metric, i.e., ES, as a more accurate choice**.

2.	**Reliable Comparisons**: Performance comparison is critical and closely related to unlearning assessment.  However, **improvement in knowledge removal typically compromise retention**, while both the goals of removal and retention are essential for effective unlearning. To overcome this dual focus, **we suggest calibrating unlearned models to align their capability of retention, thereby assessing their capabilities of removal is enough for fair comparison**.  This calibration-based framework is meaningful, **leading to many new findings in Section 6**, including the superior performance of GA-based methods over NPO after proper tuning, as well as improvements in unlearning efficacy through adjusting batch sizes and likelihood capping.

These factors have not been covered in previous works, and we are confident that our new findings hold significance for the field. below, we respond to each reviewer's valuable critiques in our individual responses. We look forward to continuing this valuable discussion during the discussion period!

---

### Meta-Review · Area_Chair_zYim · 2024-12-23

**Metareview:**

This work provides an evaluation framework for assessing LLM unlearning. The reviewers agree that the paper has made interesting conclusions and contributions, although the technical novelties and organizations are limited. It would be important for the authors to improve the paper following the reviews in the final version.

**Additional Comments On Reviewer Discussion:**

The reviewers agreed on the paper's contribution.

---

### Decision · Program_Chairs · 2025-01-22

Accept (Poster)